


# 100-kyr ice age cycles as a timescale matching problem

Takahito Mitsui[1], Peter Ditlevsen[2], Niklas Boers[3,4,5], and Michel Crucifix[5]

[1]Faculty of Health Data Science, Juntendo Univerity, Urayasu, Chiba, Japan
[2]Niels Bohr Institute, University of Copenhagen, Copenhagen, Denmark
[3]Earth System Modelling, School of Engineering & Design, Technical University of Munich, Munich, Germany
[4]Potsdam Institute for Climate Impact Research, Member of the Leibniz Association, Potsdam, Germany
[5]Earth and Life Institute, Université catholique de Louvain, Louvain-la-Neuve, Belgium

**Correspondence:** Takahito Mitsui (takahito321@gmail.com)

**Abstract.** The dominant periodicity of the late Pleistocene glacial cycles is roughly 100 kyr, rather than other astronomical periods such as 19, 23, 41, and 400 kyr. Various models explain this fact through distinct dynamical mechanisms, including synchronization of self-sustained oscillations and resonance in mono- or multi-stable systems. However, the variety of proposed models and dynamical mechanisms could obscure the essential factor for realizing the 100-kyr periodicity. We

propose the hypothesis that the ice-sheet climate system responds to astronomical forcing at the $\sim$100-kyr periodicity because the intrinsic timescale of the system is closer to 100 kyr than to other major astronomical periods. We support this idea with analyses and sensitivity studies of several simple ice age models with contrasting mechanisms.

## 1 Introduction

Glacial-interglacial cycles are a pronounced mode of climate variability in the Pleistocene, accompanied by large changes in

temperatures (Clark et al., 2024; Jouzel et al., 2007), global ice volume (Rohling et al., 2022) and greenhouse gas concentrations (Bereiter et al., 2015; Lüthi et al., 2008). Changes in global ice volume are recorded, e.g., in the oxygen isotope ratio $\delta^{18}$O of benthic foraminifera in marine sediments (Lisiecki and Raymo, 2005) (Fig. 1d), where higher $\delta^{18}$O values indicate larger ice volume and lower deep-ocean temperatures. The dominant periodicity of the late Pleistocene glacial cycles is roughly 100 kyr, as shown in its power spectral density (PSD) (Fig. 1f; see Appendix A for the PSD method).

Summer insolation in the high Northern latitudes (Fig. 1c) is supposed to be a major driver (Milankovitch, 1941; Roe, 2006) or a pacemaker (Hays et al., 1976) of the glacial cycles. It fluctuates due to long-term variations in the astronomical parameters: climatic precession $e\sin\varpi$ (and co-precession $e\cos\varpi$) with 19, 22.4 and 23.7-kyr dominant periodicities (Fig. 1b, green), obliquity $\varepsilon$ with 41-kyr periodicity (Fig. 1a), and eccentricity $e$ with 95, 124 and 405-kyr periodicities (Fig. 1b, magenta) (Laskar et al., 2004; Berger, 1978), where the periodicities of eccentricity are linked with those of climatic precession by

relations of combination tones such as $1/95 \simeq 1/23.7 - 1/19$ (Berger et al., 2005). These astronomical periodicities are in fact imprinted in the PSDs of the $\delta^{18}$O records, as shown in Fig. 1f (Hays et al., 1976).

However, despite the dominant average periodicity of the late Pleistocene glacial cycles being $\sim$100 kyr, the boreal summer insolation has only negligible power at this frequency. This discrepancy is known as the 100-kyr problem. Instead, boreal







**Figure 1.** Time series and power spectral densities (PSD) of the astronomical forcing (Laskar et al., 2004) and glacial cycles over the last 1 Myr. (a) Obliquity. (b) Climatic precession (green) and eccentricity (magenta). (c) Summer solstice insolation at $65°$N. (d) Benthic $\delta^{18}$O stack records representing glacial-interglacial cycles. The so-called LR04 record with orbital tuning (black) (Lisiecki and Raymo, 2005), the LR04 record without orbital tuning (red) (Lisiecki, 2010), and the record without orbital tuning (blue) (Huybers, 2007). Note that the vertical axis is reversed so that larger $\delta^{18}$O values, corresponding to colder conditions, are lower. (e) PSD of the eccentricity (magenta) and the PSD of the summer solstice insolation $F_{65N}$ (orange). (f) PSDs from each benthic $\delta^{18}$O record in panel e. The dashed vertical lines in (e,f) indicate major astronomical periodicities (Laskar et al., 2004).



summer insolation exhibits strong power in the 19–23.7 kyr precession band and the 41 kyr obliquity band (Fig. 1e, orange).

Henceforth, the ∼100 glacial cycles have been explained as a response to four-to-five precession cycles (Ridgwell et al., 1999; Cheng et al., 2009; Hobart et al., 2023), a response to two-to-three obliquity cycles (Huybers and Wunsch, 2005) or a combination thereof (Huybers, 2011; Tzedakis et al., 2017; Ryd and Kantz, 2024). Note that a single response to four-or-five precession cycles generally coincides with a one-to-one response to ∼100-kyr eccentricity cycles, since a deglaciation in response to climatic precession $e \sin \varpi$ tends to occur near the rising limb of eccentricity $e$ (Raymo, 1997). Thus, eccentricity

seem to impact the pace of glacial cycles via the modulation of climatic precession.

Synchronization and nonlinear resonance are two major dynamical mechanisms that result in a system's response tightly coupled with external forcing. Given their ubiquity in nature, these mechanisms are often invoked to explain the occurrence of ∼100-kyr cycles in terms of nonlinear dynamics. In *synchronization* (a.k.a. *frequency-entrainment*, *phase-locking* or *frequency-locking*)[1], the system is assumed to exhibit self-sustained oscillations in the absence of forcing, and the frequency of the under-

lying oscillations is adjusted to match one of frequencies of external forcing, its harmonics, subharmonics, or a combination of these (Pikovsky et al., 2003). Many ice age models generate ∼100-kyr cycles through the synchronization mechanism (Saltzman et al., 1984; Gildor and Tziperman, 2000; Ashkenazy and Tziperman, 2004; De Saedeleer et al., 2013; Crucifix, 2013; Ashwin and Ditlevsen, 2015; Mitsui et al., 2015; Nyman and Ditlevsen, 2019; Mitsui et al., 2023; Koepnick and Tziperman, 2024). Synchronization occurs more easily when the frequency of external forcing is closer to the natural frequency of the

system's underlying self-sustained oscillations (Pikovsky et al., 2003). Thus if the ∼100-kyr cycles are realized via synchronization, it suggests the existence of underlying self-sustained oscillations at ∼100-kyr timescale.

*Resonance*, on the other hand, refers to an enhanced output response that occurs when a system's natural frequency of oscillation matches the frequency of external forcing (Ditlevsen et al., 2020; Hagelberg et al., 1991). This term has been generalized to include a broader range of processes that involve the enhancement, suppression, or optimization of a system's

response through the variation, perturbation, or modulation of any system property (Vincent et al., 2021; Rajasekar and Sanjuan, 2016). In the *nonlinear resonance* mechanism of ∼100-kyr ice age cycles, the underlying system is commonly assumed to be either mono- or multi-stable, and the system's response to 19–23.7kyr and 41-kyr forcings is nonlinearly amplified at ∼100-kyr timescale (Ryd and Kantz, 2024), for example, at the combination tone $1/95 = 1/19 - 1/23.7$ kyr$^{-1}$ (Le Treut and Ghil, 1983). Many studies, however, use the terms of *nonlinear response* (Ganopolski, 2024; Ashkenazy and Tziperman, 2004) or

*nonlinear amplification* (Verbitsky et al., 2018), to refer to cases compatible with the generalized notion of resonance. Other types of resonance are also proposed to explain the ∼100-kyr cycles: stochastic resonance (Benzi et al., 1982; Nicolis, 1981), coherence resonance (Pelletier, 2003; Bosio et al., 2022) and vibrational resonance (Ryd and Kantz, 2024).

Despite such differences in dynamical mechanisms and underlying system types, several ice age models with distinct approaches successfully simulate proxy records with similar accuracy, reproducing the ∼100-kyr cycles. This raises an important

---

[1]In this study, we follow the definition of synchronization from Pikovsky et al. (2003), where the terms frequency entrainment, phase locking, and frequency locking are considered synonymous with synchronization, assuming the prior existence of a underlying self-sustained oscillations that is being "locked": Pikovsky et al. (2003) explicitly distinguishes these notions from resonance or nonlinear response. In many studies of glacial cycles, however, the term "phase-locking" is used to describe both synchronization and nonlinear response, regardless of the existence of underlying self-sustained oscillations.





question: if models with different mechanisms can reproduce the glacial cycles, what is the key factor that enables the $\sim$100-kyr cycles, regardless of the specific mechanism? To address this question, we examine three previously proposed ice age models, each representing a different mechanism and underlying system type: one based on synchronization, one on resonance in a mono-stable system, and one on resonance in a multi-stable system with thresholds. Through simulations changing the model's internal timescale and the amplitude of the forcing, we elucidate that the key to enabling the 100-kyr cycles is the

proximity of the intrinsic timescale of the underlying climate system to the $\sim$100-kyr periodicity of eccentricity cycles. Our results suggest that $\sim$100-kyr periodicity occurs because of the timescale matching between an astronomical timescale and one of the Earth system's intrinsic timescales.

Until now, we have referred to the dominant periodicity of the late Pleistocene glacial cycles as $\sim$100 kyr. Examining the PSDs of the benthic $\delta^{18}$O stack records over the last 1 Myr (Lisiecki and Raymo, 2005), the $\sim$100-kyr spectral peak actually

aligns with the 95-kyr eccentricity peak, and it is indeed distinct from other potential eccentricity peaks such as 124 kyr (Fig. 1e, f). Concerns could be raised about using a tuned record for such an assessment, but the same conclusion is also drawn from two other records that are free from orbital tuning (Lisiecki, 2010; Huybers, 2007). Thus, in this study, the 95-kyr periodicity is assumed as the strongest mode over the last 1 Myr (Clark et al. (2024) and Rial (1999) also specifically consider the 95-kyr periodicity). This strong imprint of the 95-kyr eccentricity periodicity (i.e., the combination tone of the climatic precession

periodicities 23.7 kyr and 19 kyr) is consistent with recent studies suggesting that the timings of deglaciations are more-tightly coupled with multiple climatic precession cycles than multiple obliquity cycles with 82 or 123 kyr (Hobart et al., 2023; Cheng et al., 2016; Abe-Ouchi et al., 2013).

The remainder of this article is organized as follows. In Section 2 we present the three simple models of ice age cycles with different mechanisms for generating $\sim$100-kyr cycles. In Section 3 the three models are analyzed to elucidate the differences

and commonality in the three mechanisms. Section 4 is devoted to the discussion. In Section 5, we conclude the article.

## 2 Models for glacial cycles

### 2.1 Self-sustained oscillator (SO) model representing the synchronization mechanism

A paradigmatic dynamical system featuring self-sustained oscillations is the oscillator of Van der Pol (1926). Crucifix and colleagues have used the forced van der Pol oscillator as a mathematical model to investigate ice age dynamics (Crucifix, 2012;

De Saedeleer et al., 2013; Crucifix, 2013). We consider a generalized version of the model:

$$\dot{x} = y + \kappa x - \frac{\mu}{3}x^3 \tag{1}$$
$$\dot{y} = -\alpha x - \beta x^3 - \theta - (\nu + \rho x)I(t) - \eta I^2(t) \tag{2}$$

with $y = \delta^{18}O - \delta - 4$, linking the variable $y$ with the ice volume proxy $\delta^{18}$O with an offset $\delta + 4$. Variable $x$ abstractly represents the 'climate' state that determines whether the system is in the glaciation or the deglaciation phase, in combination with the

insolation. It could represent the oceanic state (De Saedeleer et al., 2013), the carbon cycle, dust concentrations, or their mixed effect. Variable $I(t)$ is the standardized summer solstice insolation anomaly at 65°N, and the model's parameters are written





in Greek. The nonlinear term $\eta I^2(t)$ is included to take into account the lower sensitivity of the ice volume in the cold period (Paillard, 1998). Equations (1) and (2) contain not only the van der Pol equation, but also the equation of Duffing (1918) due to the cubic term $\beta x^3$, and the Hill equation (Magnus and Winkler, 2004) due to the multiplicative force $\rho x I(t)$ (see Appendix B

for details). Thus, it is expected to have greater flexibility to accommodate complex nonlinear oscillations than the original forced van der Pol equation.

The parameters in Eqs (1)–(2) and $\delta$ are tuned to minimize the mean squared errors between the simulated and observed $\delta^{18}$O records over the last 1 Myr (see Appendix B). The model reproduces the record of glacial cycles quite well (Fig. 2b, pink; $R = 0.88$) including the 95-kyr periodicity (Fig. S2c). For zero insolation anomaly $I(t) = 0$, the underlying system possesses

self-sustained oscillations with a periodicity of 91.7 kyr (Fig. 2b, sky blue). Such self-sustained oscillations occur over a range of insolation anomaly $-0.66 < I < 0.075$. The oscillation period varies moderately over the range $-0.66 < I < 0.075$ with a mean of about 90 kyr (Fig. S1). The internal oscillations capture the slow build up and the rapid disintegration of ice sheets. Under the astronomical forcing, the frequency entrainment occurs principally at $1/95$ kyr$^{-1}$ near the natural frequency. Equations (1)–(2) are hereafter called the Self-sustained Oscillator (SO) model.





**Figure 2.** Forced and unforced simulations of glacial cycles over the last 1 Myr: (a) Standardized summer solstice insolation at 65°N. (b) SO model with forcing (pink) and without forcing (light blue). (c) VCV18 model with forcing (violet) and without forcing (light blue). (d) G24-3 model with forcing (green) and without forcing (light blue). See Appendices A–C for the simulation settings of each model. For all three models, the corresponding scaled versions of the paleoclimatic record are shown by the black dashed line.





## 2.2 Verbitsky-Crucifix-Volobuev model representing the resonance mechanism in monostable system

Verbitsky et al. (2018) introduced a simple model of ice age cycles deduced from a scaling analysis of the governing physical laws (hereafter VCV18 model). The equations for the glaciation area ($S$), the basal temperature ($\theta$) and the ocean temperature ($\omega$) are given by

$$\dot{S} = \frac{4}{5}\zeta^{-1}S^{3/4}(a - \varepsilon I(t) - \kappa\omega - c\theta)$$

$$\dot{\theta} = \zeta^{-1}S^{-1/4}(a - \varepsilon I(t) - \kappa\omega)\{\alpha\omega + \beta[S - S_0] - \theta\}$$

$$\dot{\omega} = \gamma_1 - \gamma_2[S - S_0] - \gamma_3\omega$$

where $I(t)$ is the standardized summer solstice insolation at 65°N. The ice volume is given as $V = \zeta S^{5/4}$. See Table 1 in Verbitsky et al. (2018) for the parameter values. Since the system becomes numerically unstable near $S = 0$, we reset the $S$ value to $10^{-4}$ if it falls below $10^{-4}$. The VCV18 model can roughly simulate changes in sea level as shown in Fig. 2c (violet). Although the simulated sea level does not capture the amplitude and timing of all deglaciations (specifically, the last one), the model exhibits prominent ∼100-kyr power consistently with the record (Fig. S3c). In the absence of forcing ($I(t) \equiv 0$), it has a stable equilibrium whose Jacobian matrix has one real negative eigenvalue and a pair of complex conjugate negative real part eigenvalues. The latter defines the eigenfrequency (i.e., the *natural frequency*) of the damped oscillations. With the standard parameters, the natural periodicity of the damped oscillations is 95-kyr.

Although the astronomical forcing has major powers at ∼20-kyr and 41-kyr bands, the dominant power of the response concentrates near ∼100-kyr. Since the system does not exhibit self-sustained oscillations, the appearance of ∼100-kyr cycles in the VCV18 model can be qualified as a phenomenon of synchronization. Instead, it must be related to a nonlinear amplification of the response (Verbitsky et al., 2018), i.e., nonlinear resonance, as shown in Section 3.

### 2.3 Ganopolski model representing the resonance mechanism in multistable systems

Ganopolski (2024) discusses three simple models of ice age cycles in his Generalized Milankovitch Theory (hereafter G24-1,2,3). The G24-3 model is a model derived from ice age simulations using the Earth system model of intermediate complexity CLIMBER-2 (Calov and Ganopolski, 2005; Ganopolski and Calov, 2011; Willeit et al., 2019). The change in ice volume $v$ is defined in the glaciation- and deglaciation-regimes, respectively, as:

$$\dot{v} = \begin{cases} \dfrac{V_e - v}{t_1} & \text{if } k = 1 \text{ (glaciation regime)} \\ -\dfrac{v_c}{t_2} & \text{if } k = 2 \text{ (deglaciation regime)} \end{cases}$$

where $t_1 = 30$ kyr and $t_2 = 10$ kyr are relaxation timescales in each regime estimated from CLIMBER-2 experiments (Calov and Ganopolski, 2005). The term $V_e$ represents either of two stable equilibria depending on the 65°N summer solstice insolation anomaly $f(t)$ relative its mean over the last 1 Myr and the state $v$:

$$V_e(f) = \begin{cases} V_g(f) & \text{if } f < f_1, \text{ or } f_1 < f < f_2 \text{ and } v > V_u(f) \\ V_i & \text{if } f > f_2, \text{ or } f_1 < f < f_2 \text{ and } v < V_u(f) \end{cases}$$





where $V_g(f) = 1 + \sqrt{\frac{f_2 - f}{f_2 - f_1}}$ is the glacial equilibrium, $V_i = 0$ is the interglacial equilibrium, $f_1 \leq f \leq f_2$ is the range of multiple equilibria and $V_u(f) = 1 - \sqrt{\frac{f_2 - f}{f_2 - f_1}}$ is the unstable equilibrium separating glacial and interglacial basins (see Fig. 4 in Ganopolski (2024)). The transition from the glaciation regime ($k = 1$) to the deglaciation regime ($k = 2$) occurs if three conditions are met: $v > v_c$, $f > 0$ and $\dot{f} > 0$, where $v_c(= 1.4)$ is the critical ice volume, above which the ice sheets are likely to collapse. The transition from the deglaciation regime ($k = 2$) to the glaciation regime ($k = 1$) occurs if $f$ drops below $f_1$. Since $v$ should not be negative, we reset $v$ to 0 if it becomes negative during numerical integrations.

The G24-3 model simulates the glacial cycles well ($R = 0.82$ over 1 Myr) and has two stable equilibria for $f = 0$ (Fig. 2d). By construction, the G24-3 model does not produce self-sustained oscillations for constant insolation because its regime transitions require threshold crossings in insolation. Ganopolski (2024) mentions that the characteristic timescales of the model are $t_1 = 30$ kyr and $t_2 = 10$ kyr, and the model has no intrinsic timescale close to 100 kyr. However, the intrinsic timescale of the G24-3 model may be considered much longer than the relaxation times $t_1 = 30$ kyr and $t_2 = 10$ kyr. First, assuming the average insolation $f = 0$, the time in which the ice volume increases from $v = 0$ to the critical ice volume $v_c$ is $t_1 \ln \frac{V_g(0)}{V_g(0) - v_c} \approx 51.5$ kyr. Even after the ice volume exceeds $v_c$, the ice sheets continue to grow until the insolation anomaly $f$ changes from negative to positive. While this extra waiting time varies depending on the phase of the precession cycles, half of the precession period, approximately 10 kyr, is a reasonable expected value. Adding this value on top of 51.5 kyr, the total period from glacial inception to the onset of deglaciation is estimated as 61.5 kyr. Second, the time it takes for the ice to melt is about $t_2 = 10$ kyr. After this period of deglaciation, which usually continues during $f > 0$, the system waits for glacial inception triggered by the drop in $f$ below $f_1 = -16 \, \text{Wm}^{-2}$. This waiting time is roughly 1/4 precession cycle, i.e., $\sim 5$ kyr. The sum of the glaciation timescale and the deglaciation timescale for G24-3 model is $t_1 \ln \frac{V_g(0)}{V_g(0) - 1.5} + t_2 = 61.5$ kyr, while the timescale to complete a cycle including the extra waiting times is $t_1 \ln \frac{V_g(0)}{V_g(0) - 1.5} + t_2 + 15 = 76.5$ kyr. This timescale is closer to the 95-kyr eccentricity periodicity than other fundamental astronomical periods.

## 3 Sensitivity experiments

We conduct sensitivity experiments for the models described in Section 2 to demonstrate that the intrinsic timescale close to $\sim 100$ kyr is the key to allowing $\sim 100$ kyr periodicity for all of the three different types of models.

### 3.1 Responses to the astronomical forcing

First, we show that the three models exhibit different responses to astronomical forcing. The models are run with a scaled insolation forcing: $\tilde{I}(t) = A I(t)$ in the SO model and the VCV18 model, and $\tilde{f}(t) = A f(t)$ in the G24-3 model. The original simulations correspond to $A = 1$. The changes in the PSD for varying $A$ in steps of 0.02 are shown in Fig. 3. Specific timeseries and PSDs for $A = 0, 0.5, 1, 1.5$ and 2 are shown in Figs S2, S3 and S4.



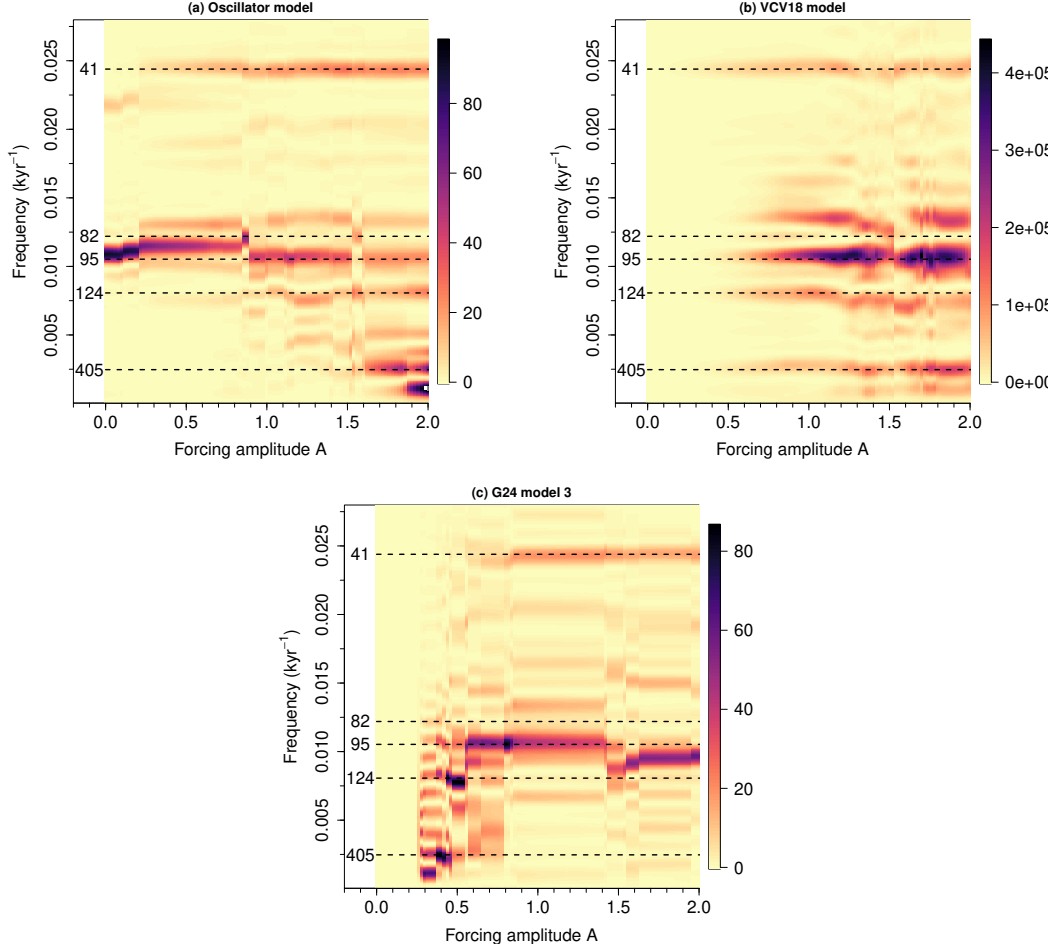

**Figure 3.** Power spectral density (PSD) for different amplitudes $A$ of the astronomical forcing: (a) SO model. (b) VCV18 model. (c) G24-3 model. The PSDs are obtained from simulations over the last 1 Myr. The magenta dashed lines indicate the major astronomical frequencies (the numbers show the corresponding periods). The precession band, 19–23 kyr, is not shown since its power is comparatively minor.

In the SO model, as shown in Fig. 3a, the PSD has its maximum at 91.7 kyr for zero forcing amplitude, $A = 0$, corresponding to self-sustained oscillations. For small $A \leq 0.84$, frequency-locking to a major astronomical periodicity is not achieved
(Fig. S2a, b). The frequency-locking to 82-kyr double obliquity periodicity is realized for a very narrow range $0.86 \leq A \leq 0.88$. The frequency locking at 95 kyr is achieved for a wide range of $0.90 \leq A \leq 1.52$. For larger $A$, the principal periodicity shifts toward the larger side, exhibiting frequency lockings to 124 kyr or 405 kyr.

In the VCV18 model shown in Fig. 3b, the total power is quite small for low $A$ since the underlying dynamics is a damped oscillation. For $A$ less than half of the original value, the PSD has a maximum at 41 kyr, although it is too small to be clearly
seen in Fig. 3b. This is simply the linear response to the 41-kyr obliquity cycles. A large power appears at the 95-kyr eccentricity





periodicity as $A$ increases to more than $0.5$. This resonance with 95-kyr eccentricity cycles is actually a nonlinear resonance to the combination tone between 19 and 23.7-kyr precession cycles. The nonlinear resonance is found near the system's natural periodicity of 95 kyr. This is consistent with the notion that the resonance typically occurs if the frequency of external forcing matches the natural frequency of the system.

In the G24-3 model shown in Fig. 3c, the power is zero for low forcing amplitude $A \leq 0.36$ because glacial inception cannot be triggered. Glacial inception is possible for $A \geq 0.38$. The main peak is located at 405 kyr for $0.38 \leq A \leq 0.42$, at 124 kyr for $0.44 \leq A \leq 0.54$, and at 95 kyr for the wide range of $0.56 \leq A \leq 1.42$. This occurs because the frequency of threshold crossing increases as $A$ increases. The principal periodicity remains close to 100 kyr for larger $A (\geq 1.44)$, but it is different from any major astronomical period.

## 175   3.2   Intrinsic timescales and responses

Next, we investigate the relationship between the principal periodicity of the output and the intrinsic timescale of the model. For this purpose, we introduce a parameter $r$ that modulates the timescale of the model, following previous studies (De Saedeleer et al., 2013; Crucifix, 2013). Each dynamical equation is scaled as $r\frac{dX}{dt} =$ r.h.s. The larger $r$, the slower the temporal variation of the model variables. In the SO model, the period of self-sustained oscillations (originally $T_0 = 91.7$ kyr) is scaled

as $rT_0$ Similarly in the VCV18 model, the natural period of the damped oscillations (originally $T_0 = 95$ kyr) becomes $rT_0$. In the G24-3 model, the intrinsic timescale for the glaciation and deglaciation is scaled as $T_{\text{int}} = r\left(t_1 \ln \frac{V_g(0)}{V_g(0) - v_c} + t_2\right) = 61.5r$ kyr. Adding the extra waiting times until the astronomical conditions are met, the timescale for forming a cycle is $T_{\text{cyc}} = r\left(t_1 \ln \frac{V_g(0)}{V_g(0) - v_c} + t_2\right) + 15 = 61.5r + 15$ kyr. The tempo of orbital forcing remains unchanged.

    We run each model by varying $(r, A) \in [0.5, 1.5] \times [0, 2]$ and measure the principal period of the simulated ice age cycles

from the PSD $S(f)$ as $T_P = 1/\text{argmax}_f S(f)$. We judge that the measured principal period $T_P$ is virtually identical to one of the major astronomical periods, $T_A$, if $|T_P - T_A| < \epsilon T_A$, where $T_A = 19, 22.4, 23.7, 41, 82, 95, 124$ or $405$ kyr (note that 82 kyr corresponds to twice the obliquity cycle). The parameter $\epsilon$ is set to be small, specifically $\epsilon = 0.028$ for the SO model and $\epsilon = 0.04$ for the VCV18 and G24-3 models. Only for the case $\epsilon = 0.04$, some $T_P$ can satisfy the condition $|T_P - T_A| < \epsilon T_A$ for both $P_A = 22.5$ kyr and $P_A = 23.7$ kyr simultaneously; in such a case, we choose the closer one to be the simulated principal

period. The results are shown in Fig. 4, as will be explained later.

    We also calculate a measure of resonance, specifically the response amplitude of signal $x(t)$ at a given frequency $f_A$ (i.e., periodicity $T_A = 1/f_A$): $Q = \sqrt{Q_s^2 + Q_c^2}$, $Q_s = \frac{2}{nT_A}\int_{-nT_A}^0 x(t)\sin(2\pi t/T_A)dt$, $Q_c = \frac{2}{nT_A}\int_{-nT_A}^0 x(t)\cos(2\pi t/T_A)dt$, where $n$ is chosen so that the integration interval spans at most the last 1000 kyr, that is $n = \lfloor 1000/T_A \rfloor$. Since the parameter $Q$ is related to the PSD as $S(f) \propto Q^2(f)$, the resonance can also be quantified by the PSD $S(f)$. However, here we employ $Q$ as

it is a widely accepted measure of resonance (Ryd and Kantz, 2024; Rajasekar and Sanjuan, 2016). The results are shown in Fig. 5.





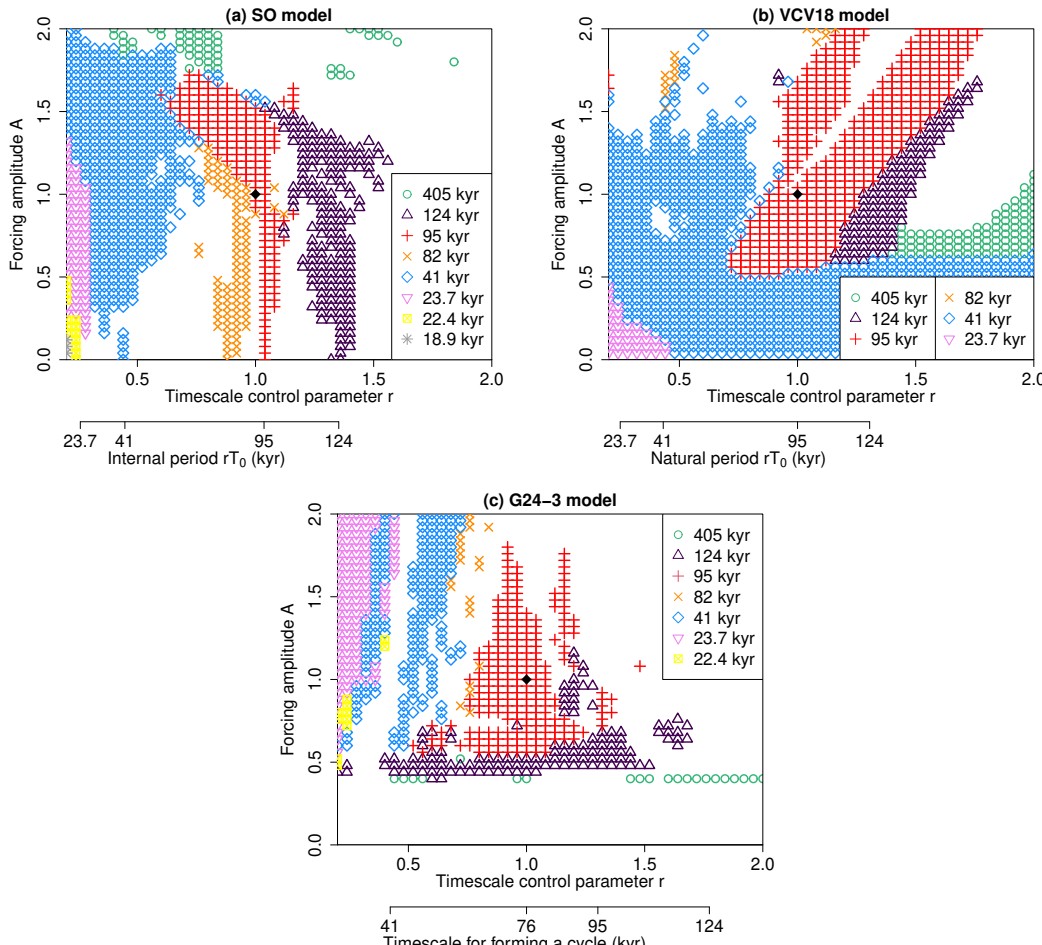

**Figure 4.** Regime diagram in $A$–$r$ space. (a) SO model (synchronization mechanism). (b) VCV18 model (nonlinear resonance mechanism in a damped oscillatory system). (c) G24-3 model (nonlinear resonance mechanism in a bistable system with thresholds). The principal period of the simulated dynamics is shown by the symbols in the legend. The most realistic simulations are obtained at $A = 1$ and $r = 1$ (black diamond). $A$ is the forcing amplitude. $r$ is the parameter controlling the timescale of the underlying system. $rT_0$ is the scaled intrinsic timescale of each model.

In the SO model shown in Fig. 4a, we find regions in which the principal period is one of the major astronomical periods. Each region comes from a point on the horizontal axis, at which the scaled internal period $rT_0$ is equal to a major astronomical period. These regions are similar to the so-called *Arnold tongues* in periodically forced systems. Within an Arnold tongue, the mean oscillation frequency, defined as the number of cycles over a large time interval, is locked to a forcing frequency or its simple rational multiple (Pikovsky et al., 2003). However, strictly speaking the regions in Fig. 4a should not be called Arnold tongues, because the principal frequency at the maximal PSD peak does not necessarily coincide with the mean oscillation frequency. We chose to call them *quasi-Arnold tongues* in Fig. 4a. Since they require only the matching between the principal





frequency and a major astronomical frequency, the quasi-Arnold tongue is a looser concept than the Arnold tongue. The quasi-
Arnold tongue corresponding to the 95-kyr periodicity is narrow and vertical (Fig. 4a). Thus, in the SO model, the 95-kyr
principal periodicity is indeed diagnosed as the system's internal frequency.

The VCV18 model does not have quasi-Arnold-tongues that touch the horizontal axis at a single point (Fig. 4b). For small
but nonzero values of $A$, the principal period is 23.7 kyr if the scaled natural period $rT_0$ is less than roughly 41 kyr, and it is
41 kyr for $rT_0 \gtrsim 41$ kyr ($0.2 \lesssim r \leq 2$). These are linear responses to the 23-kyr precession component as well as to the 41-kyr
obliquity component in the insolation forcing, respectively. For the forcing amplitude $A$ roughly above 0.5, the principal period
changes from 41 to 95 kyr for $r$ near 1. This region with the 95-kyr periodicity is sandwiched by the 41-kyr region and the
124-kyr region (Fig. 4b). This is a *nonlinear resonance tongue*, where the response amplitude with a 95-kyr periodicity is
maximized. This is confirmed by the maximum in the parameter $Q_{95}$ corresponding to 95-kyr periodicity, calculated for the
realistic forcing amplitude $A = 1$ (Fig. 5b). The nonlinear resonance tongue corresponding to 95-kyr cycles is inclined towards
the larger side of $rT_0$ as $A$ increases (Fig. 4b). This shift of natural periodicity $rT_0$ that gives the maximum amplitude (in
other words, the shift of the resonance frequency) is a characteristic of nonlinear resonance (Rajasekar and Sanjuan, 2016;
Marchionne et al., 2018). For the realistic forcing amplitude $A = 1$, the 95-kyr glacial cycles are obtained for a limited range of
natural periodicities, $83 \lesssim rT_0 \lesssim 118$ kyr. The closeness between the internal periodicity and the 95-kyr eccentricity periodicity
appears here to be key to enabling the 95-kyr dynamics also in the VCV18 model.





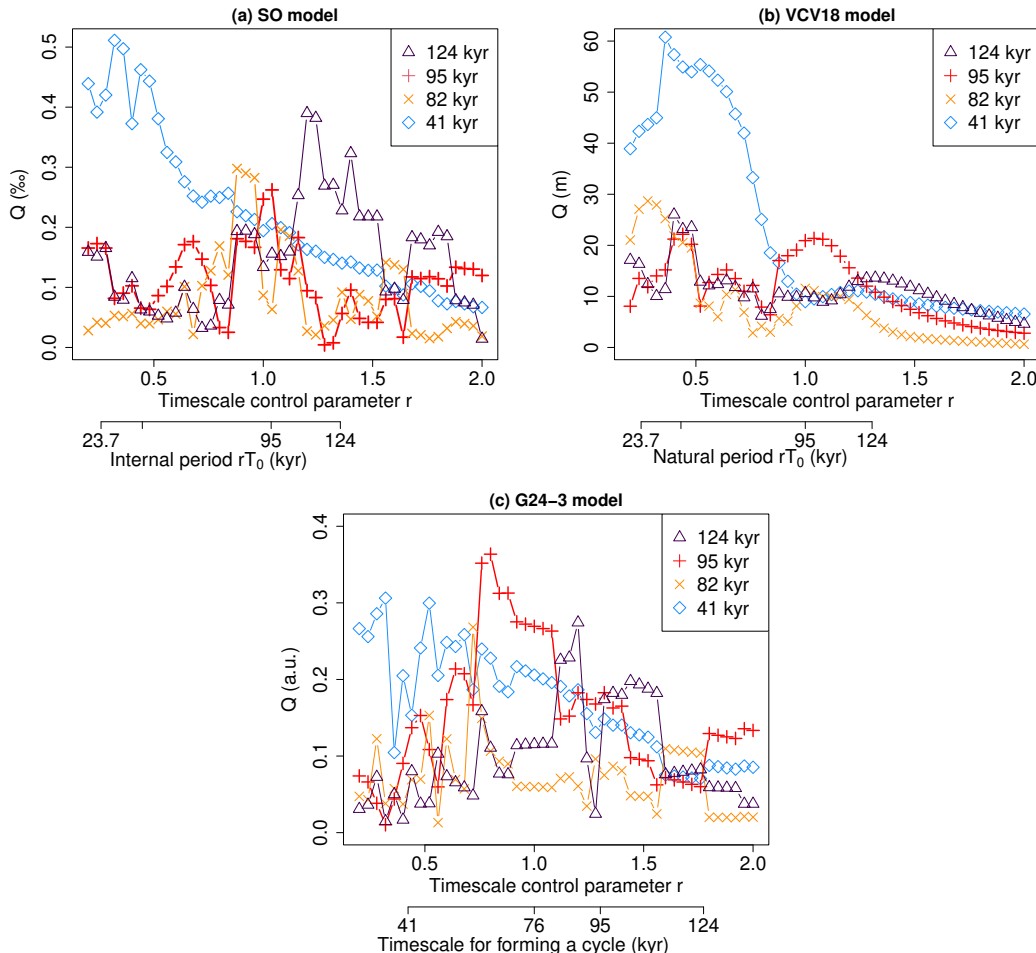

**Figure 5.** $Q$ spectrum as a function of the timescale control parameter $r$. (a) SO model (synchronization mechanism). (b) VCV18 model (nonlinear resonance mechanism in a damped oscillatory system). (c) G24-3 model (nonlinear resonance mechanism in a bistable system with thresholds). The most realistic simulations are obtained at $r = 1$, where $Q_{95}$ for the 95-kyr periodicity is maximal. $rT_0$ is the scaled intrinsic timescale of each model.

In the G24-3 model, as shown in Fig. 4c, 124-kyr glacial cycles as well as 405-kyr cycles occur for a wide range of $r$, i.e., virtually regardless of the intrinsic timescale $rT_0$. The 95-kyr cycles also occur for a wide range of intrinsic timescales $rT_0$ if $A$ is around 0.6. However, the range of $rT_0$ giving the 95-kyr cycles is limited to $66 \lesssim rT_0 \lesssim 90$ kyr at the realistic forcing amplitude $A = 1$. Thus, also in the G24-3 model, the intrinsic timescale is key to having the 95-kyr cycles. We calculate the $Q$ parameters also for this model (Fig. 5c). Among others, $Q_{95}$ corresponding to 95-kyr periodicity takes a maximum near $r = 1$. This manifests that the 95-kyr cycles in the G24-3 model are generated via nonlinear resonance.



# 4 Discussion

We have shown that three models for the late Pleistocene ice age cycles, representing the three major types of proposed mechanisms, produce ∼100-kyr cycles because they have intrinsic timescales close to ∼100-kyr. This aspect is ubiquitous across various ice age models. In fact, many models simulating 100-kyr cycles have intrinsic timescales close to ∼100 kyr as

shown in Table 1. We highlight a few examples below.

Parrenin and Paillard (2012) simulate the last 1-Myr glacial cycles particularly well using a simple model that alternates between glaciation and deglaciation regimes when astronomical parameters and ice volume exceed certain thresholds. In the model, the inherent time until the ice increases from $v_1 = 4.5$ m to $v_0 = 123$ m is $(v_0 - v_1)/\alpha_g \simeq 126$ kyr, and the timescale for deglaciation is $\tau_d \ln |(v_0 - \alpha_d \tau_d)/(v_1 - \alpha_d \tau_d)| \simeq 22$ kyr. Thus, the timescale for forming a cycle is 148 kyr. This is larger

than 100 kyr, but still closer to 100 kyr than 20, 41 and 400 kyr. In this model the astronomical forcing makes the cycles shorter than the intrinsic timescale. Benzi et al. (1982) considered a stochastic hopping in a double potential modulated by a small forcing with 100-kyr periodicity. In the model, the signal-to-noise ratio is maximal at a certain noise intensity, the so-called *stochastic resonance*. This occurs when the average waiting time between two noise-induced transitions between the two wells (the inverse of the Kramers rate) is half the forcing period, i.e., ∼50 kyr (Benzi, 2010). Therefore, the intrinsic timescale of a

cycle is $T_{\text{cyc}} = 2 \times 50 = 100$ kyr. The stochastic resonance theory is one of earliest examples treating the ∼100 kyr problem as a matching problem between the Earth's intrinsic timescale and external astronomical timescale. This theory has since been extended to align with Milankovitch theory (Matteucci, 1989; Ditlevsen, 2010). On the other hand, the piecewise linear model by Imbrie and Imbrie (1980) is the example of a model with no 100-kyr-scale intrinsic timescale, and it fails to simulate the dominant ∼100-kyr periodicity. Its intrinsic timescales are 42.5 kyr for glaciation and 10.6 kyr for deglaciation.

Nevertheless, although many models have ∼100-kyr internal timescales (Table 1), not all the models have been assessed from the viewpoint of intrinsic timescales. The existence of an underlying 100-kyr-scale intrinsic timescale is hence our *hypothesis* based on the finite set of simple models surveyed here.

In this study, we distinguished between ice age models that exhibit synchronization and those that exhibit resonance. This distinction can be subtle in some cases. (i) In synchronization theory, the forcing is generally assumed to be small relative to

the underlying self-oscillatory dynamics (Pikovsky et al., 2003). If the forcing is strong, it can significantly alter the oscillation amplitude, making it challenging to categorize the phenomenon strictly as either synchronization or resonance. (ii) *Excitable systems*, which are mono- or multistable in the absence of forcing, can produce repetitive oscillations when subject to small forcing or noise. If the frequency of such excited oscillations becomes locked to astronomical forcing, it resembles synchronization, though synchronization is typically reserved for systems with intrinsic self-sustained oscillations (Pikovsky et al.,

2003). Pierini (2023) discusses the 100-kyr cycles from the perspective of a deterministic excitation paradigm.

In Verbitsky et al. (2018) as well as Daruka and Ditlevsen (2016), the period doubling as well as the period tripling of the 41-kyr periodic cycle is proposed as the scenario to give 100-kyr-scale glacial cycles (specifically 82-kyr as well as 123-kyr cycles). This is not inconsistent with the present analysis of the VCV18 model. Indeed, if the VCV18 model is forced by the pure 41-kyr periodic forcing and if the forcing amplitude is increased, the period doubling bifurcation is observed as shown in



**Table 1.** Intrinsic timescales of models simulating ∼100-kyr glacial cycles. $T_{SO}$ is the period of self-sustained oscillations, $T_{nat}$ is the natural periodicity of damped oscillations, $T_{int}$ is the intrinsic timescale, and $T_{cyc}$ is the timescale for forming a cycle. The asterisks (*) indicate the models explored in this study.

| Model | Timescale (kyr) | Type of dynamics |
|---|---|---|
| Saltzman and Maasch (1990) | $T_{SO} = 98$ | synchronization of a sustained oscillator |
| Gildor and Tziperman (2000) | $T_{SO} \simeq 100$ | synchronization of a sustained oscillator |
| Crucifix (2012) | $T_{SO} = 103$ | synchronization of a sustained oscillator |
| Mitsui et al. (2015) | $T_{SO} = 119$ | synchronization of a sustained oscillator |
| Ashwin and Ditlevsen (2015) | $T_{SO} \simeq 100$ | synchronization of a sustained oscillator |
| Ganopolski (2024) model 1 | $T_{SO} = 101.2$ | synchronization of a sustained oscillator |
| *SO model (present study) | $T_{SO} = 91.7$ | synchronization of a sustained oscillator |
| *Verbitsky et al. (2018) | $T_{nat} = 95$ | nonlinear resonance in a damped oscillatory system |
| Benzi et al. (1982) Benzi et al. (1982); Benzi (2010) | $T_{cyc} \simeq 100$ | stochastic resonance in a bistable system |
| Parrenin and Paillard (2012) | $T_{int} \simeq 148$ | regime transitions at threshold crossings |
| *Ganopolski (2024) model 3 | $T_{cyc} = 76$ | regime transitions at threshold crossings |

Fig. S5. Comparing Fig. 4b with Fig. S5, the transition from the 41-kyr regime to the 95-kyr regime in Fig. 4b is considered an analog of a period doubling bifurcation. The true period doubling is from 41 kyr to 82 kyr. However, the 95-kyr cycles are realized instead of the 82-kyr cycles because the climatic precession forcing, which is modulated by 95-kyr eccentricity cycles, is stronger than the obliquity forcing in the power of the summer solstice insolation at 65°N (Fig. 1e).

The idea proposed here for the ∼100-kyr periodicity could potentially be extended to the 41-kyr dominant periodicity before
the Mid-Pleistocene transition (MPT) (Berends et al., 2021; Legrain et al., 2023). That is, the 41-kyr periodicity may arise if the intrinsic timescale of the climate system is close to 41 kyr. Mitsui et al. (2023) suggested this scenario by showing the 41-kyr-scale self-sustained oscillations simulated in CLIMBER-2. The G24-3 model aligns with this perspective, as it exhibits shorter timescales closer to 41 kyr before the MPT (Fig. S6), although the model does not produce self-sustained oscillations. On the other hand, how close the intrinsic timescale should be to 41 kyr depends on models. The VCV18 simulates the MPT-like
transition if the parameters are changed in time so that the positive-to-negative feedback ratio is increased (Fig. S7a). Over the last 3 Myr, the natural period of damped oscillations increases from ∼75 kyr to 95 kyr, which is calculated from the complex eigenvalue of the Jacobian matrix at the stable state (Fig. S7). Although the natural period before the MPT (75–80 kyr) is still larger than the observed 41-kyr, this subtle change is enough to obtain the 41-kyr principal periodicity before the MPT in the VCV18 model. This is already suggested by the 41 kyr region adjacent to the 95-kyr resonance tongue in Fig. 4b. While
those models suggest changes in intrinsic timescale through long-term parameter changes across the MPT, some other models produce the MPT-like periodicity change without particular parameter changes (Imbrie et al., 2011; Huybers and Langmuir, 2017; Watanabe et al., 2023). Investigating the relationship between the intrinsic timescale and the 41-kyr response using the


present approach requires comparing more models that accurately simulate the records through the MPT. We thus postpone this research to future work.

## 5   Summary

The origin of the ∼100-kyr periodicity of the late Pleistocene glacial cycles has been an enduring question in paleoclimate studies. By analyzing simple models of ice age cycles, we have demonstrated that the key factor is the proximity of the intrinsic timescale of the Earth's climate system to the ∼100-kyr periodicity of eccentricity cycles, regardless of the specific dynamical mechanism. In other words, the climate system may respond to astronomical forcing at ∼100-kyr periodicity because it is close to the intrinsic timescale of the climate system. Note that this is a hypothesis derived from a finite set of models, mostly simple ones. Investigating the intrinsic timescales of more complex models is challenging. If adjusting the timescale of a model proves difficult, artificially varying the astronomical frequencies and observing the response could be a useful approach for evaluating the validity of the timescale-matching hypothesis in complex models.

*Code and data availability.*   The R-package Palinsol is available from CRAN. The other codes used in this study will be uploaded to a Github repository after the acceptance of the paper. The tuned and untuned LR04 benthic stack records are available from https://lorraine-lisiecki. com/stack.html (last visited 2nd December 2024). The Huybers (2006) composite $\delta^{18}$O record on the depth-derived age model is available from https://www.ncei.noaa.gov/pub/data/paleo/contributions_by_author/huybers2006/huybers2006.txt (last visited 2nd December 2024).

## Appendix A: Power spectral density method

The power spectral density (PSD) $S(f)$ of a time series is estimated using the periodogram (Bloomfield, 2004), which is computed with the R function `spec.pgram` (R Core Team, 2020). By default, this function applies a split cosine bell taper to 10% of the data at both the beginning and end of the time series to minimize discontinuity effects between the start and end of the series. To increase the number of frequency bins in the periodogram, zeros are added to the end of the series to extend its length by a factor of 10 (i.e., `pad=9` in the `spec.pgram` option). Zero-padding does not fundamentally affect the PSD of the signal, but the frequency corresponding to a PSD peak is estimated with a higher resolution.

## Appendix B: van der Pol–Duffing–Hill equation

We assume that the series of glacial cycles is represented by a forced van der Pol–Duffing–Hill equation:

$$\ddot{x} + (\mu x^2 - \kappa)\dot{x} + \alpha x + \beta x^3 + \theta + (\nu + \rho x)I(t) + \eta I^2(t) = 0, \tag{B1}$$

where the parameters are written in Greek and $I(t)$ is an insolation anomaly defined below. Under the restriction to the second order differential equation, Eq. (B1) is quite comprehensive from the viewpoint of dynamical systems. First, it contains the van





der Pol equation $\ddot{x} + (\mu x^2 - \kappa)\dot{x} + \alpha x = 0$, where $\kappa$, $\nu$ and $\alpha$ are typically positive (Van der Pol, 1926; Strogatz, 2018). The van der Pol equation is well studied as a generic system showing self-sustained oscillations. Crucifix's group (Crucifix, 2012; De Saedeleer et al., 2013; Crucifix, 2013; Mitsui and Crucifix, 2016) and others (Mitsui and Aihara, 2014; Ashwin et al., 2018) have used the forced van der Pol equation as a mathematical model for investigating ice age dynamics since it can roughly fit the late Pleistocene glacial cycles.

Second, Eq. (B1) contains the Hill equation $\ddot{x} + [\alpha + \rho I(t)]x = 0$ if $I(t)$ is periodic in time (Magnus and Winkler, 2004). Furthermore, if $I(t)$ is a simple harmonic, the Hill equation is called the Mathieu equation $\ddot{x} + (\alpha + \rho \cos 2t)x = 0$. The latter is invoked to explain the rhythm of ice age cycles by Rial (1999) from the viewpoint of frequency modulation.

Third, for $\kappa < 0$, Eq. (B1) contains Duffing equation $\ddot{x} - \kappa\dot{x} + \alpha x + \beta x^3 = -\nu I(t)$ if $I(t)$ is a sinusoid. It is a paradigmatic system of nonlinear resonance as well as chaos (Duffing, 1918; Strogatz, 2018). Duffing equation exhibits forced oscillations

but not self-sustained oscillations. Dropping out the additive forcing and the nonlinear damping term, Eq. (B1) reduces to the model by Daruka and Ditlevsen (2016): $\ddot{x} + a\dot{x} - bx + bx^3 + bc - bxI(t) = 0$, where $\dot{x}$ is the global temperature anomaly, $x$ represents a climatic memory effect, and $a, b, c$ are parameters (different symbols are used in the original reference). Their model is essentially the Duffing-Hill equation since the damping term is linear. A modified version of their model can fit the proxy record well (Riechers et al., 2022).

A way to link Eq. (B1) with a proxy variable of ice age cycles is to make a first-order system taking the so-called Liénard variable $y = \dot{x} - \kappa x + \frac{\mu}{3}x^3$ (Jackson, 1989; Crucifix, 2012), which yields Eqs (1) and (2). Equation (2) links the variable $y$ with the modeled $\delta^{18}$O (‰) with an offset $\delta + 4$. The variable $x$ is an unobserved climate variable. In association with insolation forcing $I(t)$, $x$ determines whether the system is in a glaciation phase or in a deglaciation phase. The scaled summer solstice insolation anomaly $I(t)$ is defined as $I(t) \equiv (F_{65\text{N}}(t) - 495.7)/24$, where $F_{65\text{N}}(t)$ is the summer solstice insolation [Wm$^{-2}$]

at 65°N calculated with the solar constant of 1368 Wm$^{-2}$ (Fig. 1c) (Laskar et al., 2004; Crucifix, 2016). The nonlinear effect of the insolation, $\eta I^2(t)$, is included to account for the lower sensitivity of the ice volume in the cold period (Paillard, 1998). The term $-\rho x I(t)$ is a multiplicative forcing. Such a multiplicative term can appear, from physical point of view, in the energy balance via albedo effects, the ice-mass balance via temperature-precipitation feedback (Le Treut and Ghil, 1983) as well as the calcifier-alkalinity model (Omta et al., 2016).

The parameters of the equations are calibrated so as to minimize the mean squared error over the last 1 Myr. The minimization is conducted with the Nelder–Mead method implemented in R-function optim (R Core Team, 2020). The resultant parameters are $\kappa = 1.0536394044$, $\mu = 2.9662458029$, $\alpha = 0.0356079021$, $\beta = 0.0001000922$, $\theta = 0.0180996836$, $\nu = 0.0514402004$, $\rho = 0.0189082535$, $\eta = 0.0049923333$ and $\delta = 0.1801349684$.

*Author contributions.* M.C. and P.D. provided the original research plan, which was merged with another plan by T.M. and N.B. P.D.

and T.M. extended the van-der-Pol type oscillator model introduced by M.C. (Crucifix, 2012). T.M. performed the simulation and numerical analysis, with substantial contributions from the others. All authors contributed to discussing the results and analysis throughout the research. The manuscript was written by all authors, with T.M. preparing the first draft.





*Competing interests.* One of the co-authors, Michel Crucifix, is a member of the editorial board of Earth System Dynamics. The authors declare no other competing financial interests.

*Acknowledgements.* T.M. thanks Matteo Willeit for valuable discussions and Keita Tokuda for his kind support. T.M. and N.B. acknowledge funding by the Volkswagen Foundation. This is ClimTip contribution #X; the ClimTip project has received funding from the European Union's Horizon Europe research and innovation programme under grant agreement No. 101137601. N.B. acknowledges further funding by the European Union's Horizon 2020 research and innovation programme under the Marie Sklodowska-Curie grant agreement No. 956170, as well as from the Federal Ministry of Education and Research under grant No. 01LS3001A.



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
