# Peer review of "100-kyr ice age cycles as a timescale-matching problem"

_Earth System Dynamics, 2024_

## Referee Comment (RC2)

The preprint titled "100-kyr ice age cycles as a timescale matching problem" presents a compelling hypothesis that the dominant ~100-kyr periodicity of late Pleistocene glacial cycles arises from the proximity of the climate system's intrinsic timescale to the ~100-kyr eccentricity cycles. The study systematically analyzes three distinct ice age models—representing synchronization, resonance in a mono-stable system, and resonance in a multi-stable system—to demonstrate that the ~100-kyr periodicity emerges when the intrinsic timescale of the system aligns with the astronomical forcing. The manuscript is well-structured, clearly written, and addresses a long-standing question in paleoclimatology with a novel perspective. The manuscript makes a significant contribution to understanding the ~100-kyr problem by unifying diverse mechanisms under the timescale-matching hypothesis. With minor revisions—particularly expanding the discussion of the MPT and clarifying the generality of the results—the paper would be suitable for publication. I recommend acceptance after addressing the specific comments above.

The three models (SO, VCV18, G24-3) are well-chosen to represent distinct mechanisms, but their simplicity raises questions about whether the results generalize to more complex systems. For instance, how would the timescale-matching hypothesis hold in models incorporating additional feedbacks (e.g., carbon cycle, dust-albedo interactions)? A discussion on this limitation would be valuable.

The definition of "intrinsic timescale" varies across models (e.g., self-sustained oscillation period vs. relaxation timescales in bistable systems). The manuscript should clarify whether these differences affect the interpretation of timescale matching or if they represent fundamentally distinct dynamics.

The brief discussion of the MPT (Section 4) is insightful but underdeveloped. The authors suggest that the 41-kyr periodicity before the MPT could also result from timescale matching, but this is not explored in depth. Including a sensitivity analysis or model experiments addressing the MPT would significantly strengthen the paper.

The distinction between nonlinear resonance and synchronization is well-explained, but the manuscript could better highlight why this distinction matters for the  $\sim$ 100-kyr problem. For example, does the dominance of one mechanism over the other have implications for predicting future climate variability?

The power spectral density (PSD) analysis is robust, but the manuscript could include a more detailed comparison between model outputs and proxy records (e.g., time-domain metrics or phase relationships). This would help assess whether the models not only reproduce the  $\sim$ 100-kyr peak but also the timing of deglaciations.

Figures S1–S7 are cited in the text but are not included in the preprint. The authors should ensure all supplementary figures are accessible or provide descriptions in the main text.

Line 25: "Henceforth" should likely be "Previously."

Lines 70-75: It only briefly explains each chapter's general content, not the research purpose and main methods, making it hard for readers to grasp the research core at the start. Suggest the author supplement research objective and main method info. When explaining objectives, state key scientific problems to solve and expected results. When describing methods, detail model selection criteria, simulation experiment process, and data analysis methods and ideas to help readers understand the paper's core content and research context.

Line 204: The term "quasi-Arnold tongue" (Section 3.2) is introduced without a clear definition. A brief explanation or reference would aid readability.

---

## Community Comment (CC1)

**Some thoughts regarding Mitsui *et al* paper "100-kyr ice age cycles as a timescale matching problem"**

*by Mikhail Verbitsky*

**Introduction**

The authors propose "the hypothesis that the ice-sheet climate system responds to astronomical forcing at the ~100-kyr periodicity because the intrinsic timescale of the system is closer to 100 kyr than to other major astronomical periods". In terms of the similarity theory, this hypothesis suggests that the period of the system response $P$ is a function of the intrinsic timescale $\tau_{int}$ and of the amplitude and period of the astronomical forcing, $\varepsilon, T$:

$$P = \varphi(\tau_{int}, \varepsilon, T) \tag{I}$$

Since $\tau_{int}$ and $\varepsilon$ are parameters with independent dimensions then according to $\pi$-theorem:

$$\frac{P}{\tau_{int}} = \Phi\left(\frac{T}{\tau_{int}}\right) \tag{II}$$

Thus the presented paper advances a theory that the period of the system response to astronomical forcing is largely described by a similarity parameter formed by the ratio of the astronomical-forcing period to the intrinsic timescale $\frac{T}{\tau_{int}}$ ("timescale matching") and ***is independent of the amplitude of the astronomical forcing***. The numerical experiments with *phenomenological* SO and G24-3 models are mostly supportive of this theory. As it could be expected (phenomenological models are designed to produce 100-kyr periodicity), the vertically oriented strips in Fig. 4 (a, c) tell us that the period of the system response is mostly independent of forcing amplitude and is largely defined by the intrinsic timescale. At the same time - and this is what I would like to bring to the authors' attention - the results of their numerical experiments with *physical* VCV model are not consistent with the proposed theory. Even a superficial look at Fig. 4 (b) would tell us that in this case the period of the system response to astronomical forcing ***depends on the forcing amplitude***. In the following paragraphs I will try to explain why the proposed theory does not work well for ice physics.

**1. Preliminary physical considerations**

Ice-sheet physics is defined by vertical advection of ice and temperature. The timescale of this process is

$$\tau_{adv} = \frac{H}{a} \tag{1}$$

where $H$ is ice thickness and $a$ is mass influx. From scaling consideration of ice motion equations

$$H = \zeta S_0^{1/4} \tag{2}$$

where $\zeta = \left[\frac{\mu a}{(\rho g)^n}\right]^{1/(2n+2)}$, $n = 3$ is the power degree of Glen's rheological law, $\mu$ is ice viscosity, $\rho$ is ice density, $g$ acceleration of gravity, $S_0$ is ice area (Verbitsky, 1992, Bahr *et al*, 2015).

It can be seen that for all practical purposes $\zeta$ may be assumed to be a constant and since the area is involved in a power degree 1/4, the thickness of an ice sheet will not dramatically change with or without astronomical forcing. At the same time, the mass influx will change radically. Therefore the "hypothesis that the ice-sheet climate system responds to astronomical forcing at the ~100-kyr periodicity because the intrinsic timescale of the system is closer to 100 kyr than to other major astronomical periods" causes immediate concern because for the changed $a$ the intrinsic timescale may be simply irrelevant.

Let us proceed with the more rigorous reasoning.

**2. Intrinsic period of relaxation oscillations.**

We suggest that the intrinsic period of relaxation oscillations depends on ice thickness, mass influx, and the balance between positive and negative feedbacks, $V$:

$$P_{int} = \varphi(a, \zeta S_0^{1/4}, V) \tag{3}$$

If we take $a$ (m/s) and $\zeta S_0^{1/4}$ (m) as parameters with independent dimensions, then according to $\pi$-theorem:

$$\frac{P_{int}}{\tau_{int}} = \Phi(V) \tag{4}$$

$$\tau_{int} = \tau_{adv} = \frac{\zeta S_0^{1/4}}{a} \tag{5}$$

If we approximate $\Phi(V)$ as $1/(1 - V)$ implying that weak negative feedbacks (stronger $V$) provide longer relaxation periods, then

$$P_{int} = \frac{\zeta S_0^{1/4}}{a(1-V)} \tag{6}$$

For reference values of VCV model parameters, $V = 0.74$ and $P_{int} = 110$ kyr. Therefore authors' observation that the intrinsic period of VCV relaxation oscillations is close to the eccentricity period is correct.

**3. Period of the system response to orbital forcing.**

We suggest that the period of the system response, in addition to ice thickness, mass influx, and the balance between positive and negative feedbacks, depends also on the amplitude and period of the astronomical forcing, $\varepsilon, T$:

$$P = \varphi(a, \zeta S_0^{1/4}, V, \varepsilon, T) \tag{7}$$

If we take again $a$, and $\zeta S_0^{1/4}$ as parameters with independent dimensions, then

$$\frac{P}{\tau_{int}} = \Phi(\frac{\varepsilon}{a}, \frac{T}{\tau_{int}}, V) \tag{8}$$

We know from experiments with VCV model (these experiments are not part of the proposed paper, but the authors can easily replicate them) that for $T = 35 - 50$ kyr (and reference values of other parameters) the system responds with the period-doubling. This means that $\frac{P}{\tau_{int}}$ depends linearly on $T$, i.e.

$$\frac{P}{\tau_{int}} = \frac{T}{\tau_{int}} \Phi(\frac{\varepsilon}{a}, V) \text{ or}$$

$$\frac{P}{T} = \Phi(\frac{\varepsilon}{a}, V) \tag{9}$$

We can see that ***astronomical forcing makes the intrinsic timescale irrelevant.*** This does not mean though that terrestrial properties of ice-climate system have no role in ice-age periodicity, but they manifest themselves in other then $\frac{T}{\tau_{int}}$ similarity parameters, i.e., the ratio of orbital and terrestrial mass influx amplitudes $\frac{\varepsilon}{a}$ and the ratio of amplitudes of positive and negative feedbacks, $V$.

The same conclusion becomes apparent from a closer look at Fig. 4b adopted below. The timescale control parameter $r$ modifies $\zeta$ such that $\zeta' = r\zeta$ (see VCV equations), and the horizontal scale of fig. 4b has a physical meaning of relative changes of $\zeta$, i.e. $r = \zeta'/\zeta$. It can be observed that the results are independent on $\zeta$ for $0.8 \leq r \leq 1.2$ (framed below in the green rectangle). It means that, around its reference value, parameter $\zeta$ is not part of the equation (7) and the scaling law takes form of the equation (9).

[Figure]

**Fig. 4(b).** The green frame was absent in the original figure.

In other words, the period of the system response is not defined by the similarity parameter $\frac{T}{\tau_{int}}$ and therefore the VCV model does not support the hypothesis advanced by the authors. Instead, this paper provides a comprehensive support to the scaling law (9) that has been first suggested by Verbitsky and Crucifix (2020).

**4. Back to physics**

***There is no physical similarity between ice sheets with and without orbital forcing.*** Formally, it can be concluded by comparing the scaling laws (4) and (9). Physically, it may be illustrated by the following.

Sometimes parameter $\zeta$ is called "the shape factor" because it reflects ice rheology and defines the shape of an ice sheet. Since $\zeta$ ~ constant (I am not even sure that $r < 0.8$ or $r > 1.2$ are physically feasible for large ice sheets) the shape of an ice sheet remains the same with and without orbital forcing, and the mass influx is the only factor that matters for the advection timescale.

[Figure]

This may be envisioned as a fluid going through a pipe. Though the pipe is the same, there is no physical similarity between these two flows.

***The intrinsic timescale has no role in the presented results*** (around reference values) simply because the advection flow of forced ice sheet is different from the advection flow of "intrinsic" ice sheet. When the forcing amplitude is small ($A = 0.5$, other parameters being at their reference values), the negative feedbacks dominate, and the effective mass influx is about two times smaller than the reference (intrinsic) $a$. When the forcing amplitude is larger ($A = 1$, other parameters being at their reference values), positive feedbacks are engaged, and the characteristic mass influx becomes about 2 times larger than $a$, and the advection becomes much faster. Here we have interplay between faster advection and the obliquity period (i.e., obliquity-period doubling) that may resemble a non-linear resonance. Demodulation of the eccentricity period from the precession forcing (mostly done by the heat-advection equation) shifts the spectrum pick toward 95 kyr. Its closeness to the intrinsic period is just a coincidence.

**5. Conclusions**

The authors applied phenomenological thinking ($P_{int}$ has the same numerical value as $P$) to explain response of the VCV model to the orbital forcing, but phenomenology does not imply physical similarity and therefore may lead to questionable interpretations.

Nevertheless, I do believe that this paper should be published because (a) it has wealth of results that need to be studied (e.g., Fig. 4 is precious), and (b) it comprehensively exposes the difference between phenomenological (SO, G24-3) and physical (VCV) models. Having author's thoughts on what it means for ice-ages studies will be most valuable.

**References**

Bahr, D. B., Pfeffer, W. T., and Kaser, G.: A review of volume-area scaling of glaciers, Rev. Geophys., 53, 95–140, doi:10.1002/2014RG000470, 2015.

Verbitsky, M.Y.: Equilibrium ice sheet scaling in climate modeling, Climate Dynamics, 7, 105–110, https://doi.org/10.1007/BF00209611, 1992.

Verbitsky, M. Y. and Crucifix, M.: π-theorem generalization of the ice-age theory, Earth Syst. Dynam., 11, 281–289, https://doi.org/10.5194/esd-11-281-2020, 2020.

---

## Community Comment (CC2)

Dear authors,

Thank you for considering my suggestions. I think that the changes you propose move us closer to a common understanding. The following is the last iteration that hopefully will incorporate our views in one statement that makes both parties comfortable.

I (almost) agree with your observation that "in terms of $\pi$-theorem, Eq. (8) is a piecewise linear function of $T$, whose discontinuous points are determined by…(here I stumble a bit)… $\tau_{int}$". Let us consider more rigorously what governs bifurcation points.

First, we should not forget that, as I have already mentioned in my first comment, $r < 0.8$ or $r > 1.2$ may not be potentially physically feasible for large ice sheets. Nevertheless, as a purely theoretical exercise, let us expand Eq. (7), as you say, "across different modes of resonances and non-resonances".

$$P = \varphi(a, \zeta S_0^{1/4}, V, \varepsilon, T) \tag{7}$$

To better articulate my point, this time I choose parameters $\varepsilon, T$ as parameters with independent dimensions, taking Eq. (7) to the following form:

$$\frac{P}{T} = \Phi\left(\frac{\varepsilon}{a}, \frac{\varepsilon T}{\zeta S_0^{1/4}}, V\right) \tag{8}$$

During a resonance mode, as we have already established, $P \sim T$ and therefore:

$$\frac{P}{T} = \Phi\left(\frac{\varepsilon}{a}, V\right) \tag{9}$$

In a bifurcation point, the similarity parameter $\frac{\varepsilon T}{\zeta S_0^{1/4}}$ is significant. If we note that

$$\frac{\varepsilon T}{\zeta S_0^{1/4}} = \frac{T}{\tau_{int}} \frac{\varepsilon}{a} \tag{10}$$

we will arrive to the critical statement: *VCV18 bifurcation points can be described as a timescale matching problem between orbital timescale and **orbitally modified** intrinsic timescale* $\tau_{int} \frac{a}{\varepsilon} = \frac{\zeta S_0^{1/4}}{\varepsilon}$ .

This statement gives us the key for interpretation of Fig, 4(b). As I have already argued, the horizontal axis of it is $\zeta$ that can be rescaled as $\zeta S_0^{1/4}$ (these are "remains" of $\tau_{int}$). The vertical axis is $\varepsilon$. Therefore the ***slopes*** $\frac{\zeta S_0^{1/4}}{\varepsilon}$ ***are orbitally modified intrinsic timescales*** that separate nonlinear resonance tongues.

---

## Community Comment (CC3)

VCV18 model with 41−kyr periodic forcing

Bifurcation slope or orbitally modified intrinsic timescale $\tau_{int}\dfrac{a}{\varepsilon} = \dfrac{\zeta S_0^{1/4}}{\varepsilon} = \dfrac{rT_0}{A}$.
The system may bifurcate even if the intrinsic time scale is, for example, much longer than the 100-kyr eccentricity period (e.g., $r$ =2) as long as orbital forcing is strong enough (e.g., A=2).

The system is independent on intrinsic timescale and on orbitally modified intrinsic timescale (slope).

---

## Author Comment (AC1)

**Reply to Dr. Mikhail Verbitsky's comments**

Takahito Mitsui[1], Peter Ditlevsen[2], Niklas Boers[3,4,5], and Michel Crucifix[5]

[1]Faculty of Health Data Science, Juntendo Univerity, Urayasu, Chiba, Japan
[2]Niels Bohr Institute, University of Copenhagen, Copenhagen, Denmark
[3]Earth System Modelling, School of Engineering & Design, Technical University of Munich, Munich, Germany
[4]Potsdam Institute for Climate Impact Research, Member of the Leibniz Association, Potsdam, Germany
[5]Earth and Life Institute, Université catholique de Louvain, Louvain-la-Neuve, Belgium

**Correspondence:** Takahito Mitsui (takahito321@gmail.com)

Thank you very much for thoroughly reviewing our manuscript and providing very valuable feedback. Below, we reply to your comments (shown in *italic*) and propose several changes to the manuscript (shown in **bold**). We believe that these revisions will significantly enhance the quality and clarity of our work.

**Main point**

5 In your report, you argue that *astronomical forcing makes the intrinsic timescale irrelevant* and hence *the intrinsic timescale has no role in the present results* (pp. 3–4). We respectfully disagree with these statements. Your argument is based on a scaling analysis. Using Buckingham's $\pi$-theorem, the system's response period $P$ is expressed as shown in your Eq. (8):

$$\frac{P}{\tau_{\text{int}}} = \Phi\left(\frac{\varepsilon}{a}, \frac{T}{\tau_{\text{int}}}, V\right), \tag{8}$$

where $\tau_{int}$ is the system's intrinsic timescale, $\varepsilon$ is the amplitude of the forcing, $a$ is the mass influx to the ice sheets, $T$ is
10 a period of astronomical forcing, and $V$ is the parameter controlling the balance between positive and negative feedbacks. We agree with Eq. (8) itself, but you continue with the assertion that *we know from experiments with VCV model ... that for T =35–50 kyr ... the system responds with the period-doubling. This means that* $\frac{P}{\tau_{int}}$ *depends linearly on T, i.e.,*

$$\frac{P}{\tau_{int}} = \frac{T}{\tau_{int}}\Phi\left(\frac{\varepsilon}{a}, V\right), \text{ or } \frac{P}{T} = \Phi\left(\frac{\varepsilon}{a}, V\right), \tag{9}$$

*we can see that astronomical forcing makes the intrinsic timescale irrelevant.* However, Eq. (9) is only locally true in the
15 parameter space because $\frac{P}{\tau_{int}}$ depends nonlinearly on $\frac{T}{\tau_{int}}$ across different modes of resonances and non-resonances. This is shown in our Fig. 4b as a nonlinear dependence of $P$ on $\tau_{int}$ ($rT_0$ in our case). Therefore, we sustain our conclusion that the intrinsic timescale of the system is indeed critical for realizing the 100 kyr response. Of course, in each resonance mode, $P$ is fixed to $T$ or some combination of $T$s, consistently to your argument. Thus, in terms of $\pi$-theorem, Eq. (8) is a piecewise linear function of $T$, whose discontinuous points are determined by $\tau_{int}$. **In order to avoid this confusion, we clarify in the**
20 **conclusion why the intrinsic timescale $\tau_{int}$ is relevant with the system's response period $P$, while $P$ locally obeys to one of astronomical period $T$.**

**Other confusions to be clarified**

- We have not mentioned that the system's response period to the astronomical forcing *is independent of the amplitude of the astronomical forcing* (p. 1 in your report). Instead, in lines 214–216, we have mentioned that the natural periodicity leading to 95-kyr cycles shifts toward larger values as the amplitude of the astronomical forcing increases. Nevertheless, for realistic forcing amplitudes $A \approx 1$, only natural periodicities near 95-kyr, specifically $83 \leq rT_0 \leq 118$ kyr, allows to resonate with 95 kyr astronomical cycles (Fig. 4b). **In order to avoid the confusion, we mention, in the conclusion, that while the intrinsic periodicity being close to $\sim$100-kyr is necessary to realize $\sim$100-kyr cycles, the amplitude of astronomical forcing (or the sensitivity to it) is also a factor determining the response frequency.**

- You stress that *there is no similarity between ice sheets with and without forcing* (your Section 4). We also consider that ice-sheet dynamics with and without forcing are qualitatively different. In the case of the VCV18 model, the dynamics under weak focing is close to a linear response to the obliquity cycles, while the dynamics under strong forcing is characterized by nonlinear resonance at $\sim$100-kyr time scales. Our understanding is that the concept of nonlinear resonance does not entail a physical similarity between the forced and unforced systems: in general nonlinear resonance, the resonance frequency $\omega$ under the forcing and the natural frequency $\omega_0$ in the absence of forcing can differ. However, in standard cases, the nonlinear resonance frequency $\omega$ shifts continuously from the natural frequency $\omega_0$, typically following $\omega(A) = \omega_0 + \kappa A^2$, where $\kappa$ is a constant defined by the coefficient of nonlinear restoring force and the natural frequency (https://en.wikipedia.org/wiki/Nonlinear_resonance). Therefore, it is reasonable to assume that the nonlinear resonance frequency is not too far away from the natural frequency of unforced system as long as the forcing amplitude is moderate. **In the revised paper, we address the underlying assumption for discussing the nonlinear resonance in the VCV18 model based on its natural frequency in the absence of forcing.**

- The natural periodicity of the VCV18 model was calculated to be 95 kyr in our work. This value is coincidentally identical to the observed principal period of ice age cycles as well as one of the eccentricity periods. However, we do not need this coincidence for our conclusion. The natural periodicity does not have to be sharply at 95 kyr for realizing the 95 kyr cycles. Indeed, the resonance at 95 kyr can occur for a range of natural periodicities, $83 \leq rT_0 \leq 118$ kyr for the realistic astronomical forcing (see line 219). **In the revised paper, we will mention that the exact numerical match between the natural periodicity and the 95 kyr eccentricity periodicity is purely coincidental and unnecessary to achieve the 95 kyr resonance.**

Finally, we would like to thank you again. A couple of equations introduced in your report help theoretical considerations. While your comments are critical, we believe that apparent contradictions between your opinions and our thoughts can be solved by careful clarifications proposed above.

---

## Author Comment (AC2)

**Reply to the comments by Dr. Holger Kantz (referee)**

Takahito Mitsui[1], Peter Ditlevsen[2], Niklas Boers[3,4,5], and Michel Crucifix[5]

[1]Faculty of Health Data Science, Juntendo Univerity, Urayasu, Chiba, Japan
[2]Niels Bohr Institute, University of Copenhagen, Copenhagen, Denmark
[3]Earth System Modelling, School of Engineering & Design, Technical University of Munich, Munich, Germany
[4]Potsdam Institute for Climate Impact Research, Member of the Leibniz Association, Potsdam, Germany
[5]Earth and Life Institute, Université catholique de Louvain, Louvain-la-Neuve, Belgium

**Correspondence:** Takahito Mitsui (takahito321@gmail.com)

Thank you very much for reviewing our manuscript in detail and giving us very useful feedback. Below, we reply to your comments and questions (shown in *italic*), and propose several changes to the manuscript (shown in **bold**). We believe that these revisions will enhance the quality and clarity of our work.

**Main issue**

5 *I missed (or may have overlooked the discussion of) only one aspect in this issue of the 100 kyr cycles: The lack of spectral power at 100 kyr in the 65N insolation time series means that the driving signal lacks this frequency component. Nonetheless they state in line 60 that 'proximity of the intrinsic time scale .... to the 100 kyr periodicity of the eccentricity cycles' is relevant, i.e., they consider the 100 kyr period of the driver to be due to eccentricity. This seems to be in contradiction to the fact that in the PSD of 65N insolation there is no enhanced power in this frequency band, and they also cite Berger who proposed a kind*
10 *of beating frequency of the 23.7 and 19 kyr modes to be responsible for the 100 kyr cycle.*

Thank you for pointing out this aspect. The apparent contradiction is resolved as follows: The Earth system does not simply respond to the precession cycles but responds to the beat frequency generated by the addition of the 23.7- and 19-kyr precession cycles. The beat frequency is strictly equal to the frequency of the 95-kyr eccentricity cycles (cf. $1/19 - 1/23.7 = 1/95$). Thus, the nonlinear, subharmonic-type, response to the 23.7- and 19-kyr precession cycles is physically similar to a response to the
15 95-kyr eccentricity cycles. In the Introduction of our discussion paper (lines 27–30), we have briefly mentioned the above fact. **However, in the revised paper, we will make the corresponding text clearer and will mention the above solution to the apparent contradiction again in the summary paragraph**.

*The fact that the eccentricity period of 95 kyr is close to the 100 kyr, is this essential or just by chance? Perhaps the authors*
20 *can comment on this.*

We consider the so-called 100-kyr cycles to be a simplified characterization of the ice age cycles, whose mean periodicity is closer to 95 kyr, as observed in the power spectra of the records (Fig. 1). Thus, there is no exact 100-kyr cycles.[1] **In the revised**
* * *
[1]The eccentricity has also a periodicity of 98. 857 kyr [Laskar et al. 2005] (or 99.590 kyr in Berger et al. 2005), which is closer to 100 kyr than 95 kyr. However, the power of the 98. 857 kyr cycles is much less than that of 95 kyr cycles.

**manuscript, we state that while the ice age cycles are generally described as having a roughly 100-kyr periodicity, they may be more closely associated with the 95-kyr eccentricity cycles.**

25   **Other minor issues**

*Line 25, "Hencefore, the ≈ 100 glacial cycles...": kyr is missing.*
It was a typo. **We will add '-kyr'.**

*Line 117: "... the VCV18 model CANNOT be qualified as ... synchronization"?*
30   It was our mistake. **We will change 'can' to 'cannot'.**

*Line 155: What is the difference between $I(t)$ and $f(t)$? In line 86 it is said "$I(t)$ is the standardized summer solstice insolation anomaly at 65N", as well as in line 107. $f(t)$ is defined in line 128 as '65N summer solstice insolation anomaly'. Perhaps the authors can invest one more line to clarify this (also where the mean over the past 1 Myr appears and what $f_1$, $f_2$ are).*
35   Thank you for pointing out this. $I(t)$ is the standardized anomaly scaled by its standard deviation, and $f(t)$ is just an anomaly NOT scaled by its standard deviation. **In the revised paper, we will add the following sentence: "Note that $f(t)$ is an anomaly that is not scaled by its standard deviation, different from $I(t)$ in the previous two models." We also clarify the critical insolation anomalies $f_1$ and $f_2$, between which the system has two glacial and interglacial attractors.**

40   We would like to thank you again for your thoughtful comments and very useful feedback.

---

## Author Comment (AC3)

**Reply to Dr. Mikhail Verbitsky's comments (CC2 and CC3)**

Takahito Mitsui[1], Peter Ditlevsen[2], Niklas Boers[3,4,5], and Michel Crucifix[5]

[1]Faculty of Health Data Science, Juntendo Univerity, Urayasu, Chiba, Japan
[2]Niels Bohr Institute, University of Copenhagen, Copenhagen, Denmark
[3]Earth System Modelling, School of Engineering & Design, Technical University of Munich, Munich, Germany
[4]Potsdam Institute for Climate Impact Research, Member of the Leibniz Association, Potsdam, Germany
[5]Earth and Life Institute, Université catholique de Louvain, Louvain-la-Neuve, Belgium

**Correspondence:** Takahito Mitsui (takahito321@gmail.com)

Thank you very much for your further comments, which are very thoughtful and valuable for us. We are glad that we are coming closer to common understanding through discussions.

In your report CC2, you state that "*VCV18 bifurcation points can be described as a timescale matching problem between orbital timescale and orbitally modified intrinsic timescale*". We agree with your point that the range of the intrinsic timescale allowing a particular resonance ($P \sim T$) depends on the amplitude of astronomical forcing, and that such resonances may not occur if the forcing is too weak. On that basis, we enlighten the fact that, in many ice age models under astronomical forcing **with a realistic amplitude**, the $\sim$100-kyr responses arise if the model's intrinsic timescale is close to $\sim$100 kyr. That is, our conclusion sustains for the realistic amplitude of the astronomical forcing: $A \approx 1$ in our terminology and $\varepsilon \approx 1$ in VCV18's term.

Inspired by your scaling analysis, we propose the following physical argument. It does not use the Pi-theorem but as we show next it converges to a conclusion similar to yours. Following the VCV18 paper, the height of the fully developed ice sheet is given by $H = \zeta S_0^{1/4}$ and the snow accumulation rate is $a$. Using your comment (CC1), the intrinsic time scale of advection **in the absence of forcing** is given as

$$\tau_{adv} = \frac{H}{a} = \frac{\zeta S_0^{1/4}}{a}.$$

Since the snow accumulation rate $a$ and the forcing term $\varepsilon F_S(t)$ appear as $a - \varepsilon F_S(t)$ in the dynamical equations of VCV18, we assume that the **net ice accumulation rate under a forcing cycle** scales as $a - c\varepsilon$: this is similar to $a - \varepsilon F_S(t)$ but we introduce a cycle-specific coefficient $c$. Indeed, the astronomical forcing is a complicated signal and its amplitude from cycle to cycle. For this cycle (of period, say, $T$) to actually entertain a resonance with glaciation dynamics, we expect the typical ice build-up time to match $T$, i.e.,

$$(\text{glaciation period}) = \frac{(\text{maximal ice-sheet height})}{(\text{net ice accumulation rate})} = \frac{H}{a - c\varepsilon} \sim T.$$

$$\Leftrightarrow \; c \sim \frac{1}{\varepsilon}\left(a - \frac{H}{T}\right) = \frac{a}{\varepsilon T}\left(T - \tau_{adv}\right).$$

This equation must hold for a majority of cycles, that is, for a range of $c$ denoted by $-c_1 < c < c_2$ ($c_1, c_2 > 0$). Thus,

$$-c_1 \lesssim \frac{a}{\varepsilon T}\left(T - \tau_{adv}\right) \lesssim c_2$$

[Figure]

**Figure 1.** The parameter region derived from the simple physical consideration. The resonance with the astronomical period $T$ is possible at least within the triangular region.

That is,

$$\varepsilon \gtrsim \frac{a}{c_1 T}\left(\tau_{adv} - T\right) \ \text{and} \ \varepsilon \gtrsim \frac{a}{c_2 T}\left(T - \tau_{adv}\right).$$

These inequalities imply a triangular region in $\tau_{adv}$–$\varepsilon$ space (Fig. 1 here). The system may resonate at the astronomical period $T$ at least within the triangular region. If we interpret $\tau_{adv}$ as the intrinsic timescale of the system and if use the notation of our article ($\tau_{adv} = rT_0$ and $\varepsilon = A$), the above inequalities are

$$A \gtrsim \frac{a}{c_1 T}\left(rT_0 - T\right) \ \text{and} \ A \gtrsim \frac{a}{c_2 T}\left(T - rT_0\right).$$

The resonance may occur at least within this region, but the actual resonance region is more complicated than suggested from the above equations because of nonlinear effects (cf. Figs. 4 and S5 in our article).

The inequalities derived here are slightly different from what you drive using a scaling analysis in CC2. However, we reach essentially the same conclusion that the range of the intrinsic timescale leading to a particular resonance ($P \sim T$) must depend on the forcing amplitude if the forcing amplitude changes significantly. Our conclusion holds for the astronomical forcing with realistic amplitude. **This point will be addressed in the revised manuscript.**

We would like to thank you again for guiding us to the physical considerations.

---

## Author Comment (AC4)

**Reply to the comments by Referee #2**

Takahito Mitsui1, Peter Ditlevsen2, Niklas Boers3,4,5, and Michel Crucifix5 1Faculty of Health Data Science, Juntendo Univerity, Urayasu, Chiba, Japan 2Niels Bohr Institute, University of Copenhagen, Copenhagen, Denmark 3Earth System Modelling, School of Engineering & Design, Technical University of Munich, Munich, Germany 4Potsdam Institute for Climate Impact Research, Member of the Leibniz Association, Potsdam, Germany 5Earth and Life Institute, UCLouvain, Louvain-la-Neuve, Belgium

Correspondence: Takahito Mitsui (takahito321@gmail.com)

Thank you very much for reviewing our manuscript in detail and giving us very useful feedback. We needed some time to thoroughly consider your comments before responding. Below, we have listed your questions and comments (*italicized*), followed by our responses and proposed revisions to the manuscript (shown in violet). We believe that these revisions will enhance the quality and clarity of our work.

5

The three models (SO, VCV18, G24-3) are well-chosen to represent distinct mechanisms, but their simplicity raises questions about whether the results generalize to more complex systems. For instance, how would the timescale-matching hypothesis hold in models incorporating additional feedbacks (e.g., carbon cycle, dust-albedo interactions)? A discussion on this limitation would be valuable.

10 Indeed, our numerical investigations and survey in Table 1 focus on simple models. This is a limitation of the present work. Nevertheless, as mentioned below, some studies in the literature offer insights on how our timescale-matching hypothesis may hold in more complex models.

First, an early study by Oerlemans (1982) demonstrated that an ice-sheet–bedrock system could exhibit 100-kyr-scale selfsustained oscillations especially due to strong feedbacks involving basal melting and sliding of the ice sheets. This model is an

15 instance that the 100-kyr-scale intrinsic oscillations are relevant for producing 100-kyr cycles under insolation forcing, even though our knowledge of lithosphere physics has since been revised.

Second, since the G24-3 model was, according to its author, inspired by experiments using the Earth system model of intermediate complexity, CLIMBER-2 model (Ganopolski, 2024). If we follow this argument, our results obtained from the G24-3 model can be relevant with complex climate systems including carbon cycles and dust–albedo interactions.

20 Third, and perhaps more importantly, Mitsui et al. (2023) showed that a version of the CLIMBER-2 model exhibits selfsustained oscillations with periodicities of several hundred thousand years, due to the glaciogenic dust feedback and carbon cycle feedbacks. Such long timescales are crucial for ~100-kyr ice age cycles simulated in the CLIMBER-2 model under the forcing. These previous findings support the timescale-matching hypothesis proposed in this study. In the Discussion section of the

**25 revised manuscript, we will address the limitation and the above supports from complex models.**

The definition of "intrinsic timescale" varies across models (e.g., self-sustained oscillation period vs. relaxation timescales in bistable systems). The manuscript should clarify whether these differences affect the interpretation of timescale matching or if they represent fundamentally distinct dynamics.

30 Yes, the differences in model dynamics do affect the interpretation of timescale matching:

If the underlying model exhibits self-sustained oscillations, the period of glacial cycles under forcing tends to be tightly coupled with the period of the internal oscillation, as shown by the nearly vertical quasi-Arnold tongues in Fig. 4a. In such cases, the observed  $\sim$ 100-kyr cycles may suggest the presence of self-sustained oscillations with a period very close to 100 kyr. In contrast, in systems with nonlinear resonance involving damped oscillations, the resonance frequency can deviate from the

as natural frequency of damped oscillations depending on the amplitude of the forcing. The importance of this deviation has been emphasized in the community comments by Dr. Verbitsky. In the case of the bistable model (G24-3), the intrinsic timescale is not purely internal, but instead includes a  $\sim$ 10-kyr-scale waiting time until favorable astronomical conditions are met.

Considering these differences, we suggest viewing the intrinsic timescale as the time required to form a cycle. Notably, many models that reproduce realistic 100-kyr glacial cycles tend to exhibit such intrinsic timescales close to 100 kyr. These observa-

40 tions support the generality of the timescale-matching hypothesis proposed in this study. In the the revised manuscript, we clarify the definition of intrinsic timescales across models introducing a schematic figure, and then discuss the different interpretations of timescale matching in each type of model.

The brief discussion of the MPT (Section 4) is insightful but underdeveloped. The authors suggest that the 41-kyr periodicity
before the MPT could also result from timescale matching, but this is not explored in depth. Including a sensitivity analysis or model experiments addressing the MPT would significantly strengthen the paper.

We thank the reviewer for this suggestion. It is indeed possible to extend our results into the pre-MPT period, although we had planned it for future work. In order to make our results more comprehensive, we will expand the present discussion on the MPT in the revised manuscript. Specifically, we will present sensitivity experiments in the 41-kyr world before the
50 MPT (i.e., a new figure corresponding to Fig. 4).

The distinction between nonlinear resonance and synchronization is well-explained, but the manuscript could better highlight why this distinction matters for the 100-kyr problem. For example, does the dominance of one mechanism over the other have implications for predicting future climate variability?

55 Nonlinear resonance and synchronization are distinct concepts in nonlinear science, and using these terms properly is important to avoid confusion in discussions of ice age cycles. However, the present study does not aim to determine which mechanism is most plausible, thus the most adequate for long-term predictions. Our objective was to provide a unified perspective on these mechanisms through the lens of timescale matching. As you suggest, if one plausible mechanism is ultimately identified, it may have implications for predicting future climate 60 variability. For example, if the Earth system exhibits self-sustained oscillations, it may have a tendency to enter a new ice age spontaneously over the next tens of thousands of years. We keep in mind that in reality, anthropogenic forcing plays a significant role, and any such prediction must clearly be verified using more realistic models.

The power spectral density (PSD) analysis is robust, but the manuscript could include a more detailed comparison between 65 model outputs and proxy records (e.g., time-domain metrics or phase relationships). This would help assess whether the models 65 not only reproduce the 100-kyr peak but also the timing of deglaciations.

Yes, the time-domain metrics or phase relationships are definitely useful for assessing models with respect to the timings of deglaciations. The timings of deglaciations themselves are however complicated metrics because the last 800-kyr contains eleven deglaciations. The most simple time-domain metric would be the Pearson's correlation coefficient (PCC) between model

- 70 outputs and the proxy record would. The PCC evaluates phase relationships, although it does not focus on deglaciations. In the revised manuscript, we will include a new supplementary figure demonstrating that timescale matching is also necessary to achieve a high correlation with the data, using the PCC as the metric. On the other hand, we cannot exclude the possibility that even good models may fail to reproduce the precise timing of deglaciations, particularly if that timing is sensitively dependent on parameters or influenced by stochastic forcings (Crucifix, 2013; Mitsui and Aihara, 2014; Mitsui et al.,
- 75 2015; Mitsui and Crucifix, 2016). Therefore, caution is warranted when using correlation-based metrics for model comparison.

Figures S1–S7 are cited in the text but are not included in the preprint. The authors should ensure all supplementary figures are accessible or provide descriptions in the main text.

It is unfortunate that you could not access the supplementary figures during the review. Actually they have been provided in the preprint page at: https://esd.copernicus.org/preprints/esd-2024-39/esd-2024-39-supplement.pdf. In the revised manuscript, we ensure that the manuscript includes the link to the supplementary material.

Line 25: "Henceforth" should likely be "Previously."

90

Indeed, "Henceforth" is not suitable in this context. Instead, we find that "hence" provides a clearer logical connection between the sentences. Accordingly, we have rephrased the text as follows: Hence, the ~100-kyr glacial cycles have previously been explained as ...

Lines 70-75: It only briefly explains each chapter's general content, not the research purpose and main methods, making it hard for readers to grasp the research core at the start. Suggest the author supplement research objective and main method info. When explaining objectives, state key scientific problems to solve and expected results. When describing methods, detail

model selection criteria, simulation experiment process, and data analysis methods and ideas to help readers understand the paper's core content and research context.

We acknowledge that the research purpose and main methods may not have been clearly conveyed, possibly because they were embedded mid-paragraph (lines 54–60).

95 In response to your comment, we will revise the final paragraph of the Introduction to clearly state the objectives of the paper, outline the key scientific questions addressed, and describe the main methods, including model selection criteria, simulation procedures, and data analysis approaches.

*Line 204: The term "quasi-Arnold tongue" (Section 3.2) is introduced without a clear definition. A brief explanation or reference would aid readability.*

The original Arnold tongues are triangular-shaped regions in parameter space where synchronization occurs (Pikovsky et al., 2003). In this study, we refer to quasi-Arnold tongues as triangular regions in which the principal frequency of a self-sustained oscillator under external forcing matches either one of the forcing frequencies or a linear combination of them. This concept extends the classical definition of Arnold tongues to cases where exact frequency locking is replaced by more flexible frequency entrainment patterns.

We would like to thank you again for your thoughtful comments and very useful feedback.

105

**References**

Crucifix, M.: Why could ice ages be unpredictable?, Climate of the Past, 9, 2253–2267, 2013.

- 110 Ganopolski, A.: Toward Generalized Milankovitch Theory (GMT), Climate of the Past, 20, 151–185, 2024.
  - Mitsui, T. and Aihara, K.: Dynamics between order and chaos in conceptual models of glacial cycles, Climate dynamics, 42, 3087–3099, 2014.
    - Mitsui, T. and Crucifix, M.: Effects of additive noise on the stability of glacial cycles, Mathematical Paradigms of Climate Science, pp. 93–113, 2016.
- 115 Mitsui, T., Crucifix, M., and Aihara, K.: Bifurcations and strange nonchaotic attractors in a phase oscillator model of glacial-interglacial cycles, Physica D: Nonlinear Phenomena, 306, 25–33, 2015.
  - Mitsui, T., Willeit, M., and Boers, N.: Synchronization phenomena observed in glacial-interglacial cycles simulated in an Earth system model of intermediate complexity, Earth System Dynamics, 2023, 1277–1294, https://doi.org/10.5194/esd-14-1277-2023, 2023.
    Oerlemans, J.: Glacial cycles and ice-sheet modelling, Climatic Change, 4, 353–374, 1982.
- 120 Pikovsky, A., Kurths, J., Rosenblum, M., and Kurths, J.: Synchronization: a universal concept in nonlinear sciences, 12, Cambridge university press, 2003.

---

## Author Response (AR2)

Dr. Takahito Mitsui
Faculty of Health Data Science
Juntendo University, Japan
takahito321@gmail.com

8 July 2025

Dear Prof. Claudia Pasquero,

Thank you very much for reviewing our manuscript entitled "100-kyr ice age cycles as a timescale-matching problem." We apologize for the delay in our resubmission and appreciate your kindness in allowing the review process to continue.

We are pleased to resubmit a revised version of the manuscript, which has been updated in accordance with the comments from you and the referees. Below, you will find a summary of the main changes as well as our detailed, point-by-point responses to the comments. We believe that the revisions significantly improve the clarity and overall quality of the manuscript.

**Summary of the main changes**

- Following the comments from the editor as well as Referee #2, we have added a paragraph in the Discussion section (lines 264–274) that supports our timescale-matching hypothesis by referencing previous studies using a physical ice-sheet model and an Earth system model of intermediate complexity (CLIMBER-2).

- We have clarified the terminology related to the various timescales used in the manuscript–such as intrinsic timescales, the natural period, and the timescale for forming a cycle–and revised the manuscript accordingly. In particular, we have added a new Fig. 3 to summarize these timescales.

- In response to a comment by Referee #2, we have included a new Supplementary Fig. S5, which demonstrates that the timescale match is also necessary to achieve a high correlation with the data.

- Following the reviewers' and public comments below, we have revised several parts of the text to make the sentences and technical terms clearer.

In what follows, the comments we received are shown in *italics*, and our proposed revisions to the manuscript are highlighted in **bold**.

**Reply to Editor's comment**

*Based on the overall positive evaluation of the manuscript by the referees and on the answers the authors provided to the raised concerns, I believe that a revision will likely significantly improve the manuscript. As the authors propose, clarifications to specific points will help the readers to correctly interpret the results presented and further discussion will be valuable in better focusing on the physical processes relevant for the problem under study. I thus encourage the authors to submit a revised version of the manuscript, taking into account all concerns raised by the reviewers.*

We are grateful for your helpful suggestion regarding the revision. In particular, you encouraged us to focus on the physical processes relevant to the problem under study. Accordingly, we have added a paragraph in the Discussion section (lines 264–274) that supports our timescale-matching hypothesis by referencing previous studies using a physical ice-sheet model and an Earth system model of intermediate complexity (CLIMBER-2).

At the same time, while hundreds of studies have investigated the physical processes behind glacial–interglacial cycles, our goal is not to determine which explanation is correct or most relevant. Rather, we aim to use mathematical reasoning to extract a general property shared by these various models and hypotheses. We believe this constitutes a novel aspect of the present study.

**Reply to Referee #1 (Dr. Holger Kantz)**

**Main issue**

*I missed (or may have overlooked the discussion of) only one aspect in this issue of the 100 kyr cycles: The lack of spectral power at 100 kyr in the 65N insolation time series means that the driving signal lacks this frequency component. Nonetheless they state in line 60 that 'proximity of the intrinsic time scale .... to the 100 kyr periodicity of the eccentricity cycles' is relevant, i.e., they consider the 100 kyr period of the driver to be due to eccentricity. This seems to be in contradiction to the fact that in the PSD of 65N insolation there is no enhanced power in this frequency band, and they also cite Berger who proposed a kind of beating frequency of the 23.7 and 19 kyr modes to be responsible for the 100 kyr cycle.*

Thank you for pointing out this aspect. The apparent contradiction is resolved as follows: The Earth system does not simply respond to the precession cycles but mainly responds to the beat frequency generated by the addition of the 23.7- and 19-kyr precession cycles. The beat frequency is strictly equal to the frequency of the 95-kyr eccentricity cycles (cf. $1/19 - 1/23.7 = 1/95$). Thus, the nonlinear, subharmonic-type, response to the 23.7- and 19-kyr precession cycles is physically similar to a response to the 95-kyr eccentricity cycles. In the Introduction of our discussion paper (lines 27–30), we have briefly mentioned the above fact. Moreover, in the revised paper, we have made the corresponding text clearer and have mentioned the above solution to the apparent contradiction again in the summary paragraph (lines 328–331): **"Although the astronomical forcing possesses only negligible power at 100-kyr-band, these models exhibit ∼100-kyr ice age cycles as a response to the amplitude-modulation of climatic precession cycles. This is physically equivalent with the response to ∼100-kyr eccentricity cycles that modulate the amplitude of climatic precession."**

*The fact that the eccentricity period of 95 kyr is close to the 100 kyr, is this essential or just by chance? Perhaps the authors can comment on this.*

We consider the so-called 100-kyr cycles to be a simplified characterization of the ice age cycles, whose mean periodicity is closer to 95 kyr, as observed in the power spectra of the records (Fig. 1f). Thus, there is no exact 100-kyr cycles.[1] In the revised manuscript (lines 32–41), we have stated that while the ice age cycles are generally described as having a roughly 100-kyr periodicity, they may be more closely associated with the 95-kyr eccentricity cycles. In this work, we have proposed the hypothesis that the Earth system responds most strongly to the 95-kyr eccentricity cycles since the intrinsic timescale is close to it.
* * *
[1] The eccentricity has also a periodicity of 98. 857 kyr [Laskar et al. 2005] (or 99.590 kyr in Berger et al. 2005), which is closer to 100 kyr than 95 kyr. However, the power of the 98.857 kyr cycles is much less than that of 95 kyr cycles.

**Other minor issues**

It was a typo. We have added '-kyr'.

It was our mistake. We have changed 'can' to 'cannot'.

$I(t)$ is the standardized anomaly scaled by its standard deviation, and $f(t)$ is just an anomaly NOT scaled by its standard deviation. In the revised paper, we have added the following sentence (lines 139-140): **"Note that $f(t)$ is an anomaly that is not scaled by its standard deviation, different from $I(t)$ in the previous two models."** $f_1 = -1.6 \ \mathrm{Wm}^{-2}$ and $f_2 = 1.6 \ \mathrm{Wm}^{-2}$ are the critical insolation anomalies, between which the system has two glacial and interglacial attractors, and are now described in line 137.

**Reply to Referee #2**

Indeed, our numerical investigations and survey in Table 1 focus on simple models. This is a limitation of the present work. Nevertheless, as mentioned below, some studies in the literature offer insights on how our timescale-matching hypothesis may hold in more complex models.

First, an early study by Oerlemans (1982) demonstrated that an ice-sheet–bedrock system could exhibit 100-kyr-scale self-sustained oscillations especially due to strong feedbacks involving basal melting and sliding of the ice sheets. This model is an instance that the 100-kyr-scale intrinsic oscillations are relevant for producing 100-kyr cycles under insolation forcing, even though our knowledge of lithosphere physics has since been refined.

Second, since the G24-3 model was, according to its author, inspired by experiments using the Earth system model of intermediate complexity, CLIMBER-2 model (Ganopolski, 2024). If we follow this argument, our results obtained from the G24-3 model can be relevant with complex climate systems including carbon cycles and dust–albedo interactions.

Third, and perhaps more importantly, Mitsui et al. (2023) showed that a version of the CLIMBER-2 model exhibits self-sustained oscillations with periodicities of several hundred thousand years, due to the glaciogenic dust feedback and carbon cycle feedbacks. Such long timescales are crucial for ~100-kyr ice age cycles simulated in the CLIMBER-2 model under the forcing.

These previous findings support the timescale-matching hypothesis proposed in this study. In the Discussion section of the revised manuscript (lines 264–274), we have addressed the limitation and the above supports from complex models.

*The definition of "intrinsic timescale" varies across models (e.g., self-sustained oscillation period vs. relaxation timescales in bistable systems). The manuscript should clarify whether these differences affect the interpretation of timescale matching or if they represent fundamentally distinct dynamics.*

Yes, the different dynamical mechanisms leading to ∼100-kyr cycles affect how timescale matching hypothesis should be interpreted. We have added a new paragraph to explain the differences in interpretation in the revised manuscript (lines 278–286). In Introduction, we have described the distinction between synchronization and resonance, but we have also mentioned some similarity between them in Discussion (lines 278–295).

*The brief discussion of the MPT (Section 4) is insightful but underdeveloped. The authors suggest that the 41-kyr periodicity before the MPT could also result from timescale matching, but this is not explored in depth. Including a sensitivity analysis or model experiments addressing the MPT would significantly strengthen the paper.*

We thank the reviewer for this valuable suggestion. In the interactive discussion, we proposed to extend our results to the pre-MPT period and indeed conducted the sensitivity analysis. Figure 1 below presents an extension of our sensitivity experiments to the 41-kyr world before the MPT. This result supports that the timescale matching also holds in 41-kyr world: the 41-kyr dynamics occurs in a limited range of the scaled intrinsic time scale near 41 kyr in panels (a) and (d). The region of 41-kyr dynamics is bounded from the lower side in (c) and (d). However, during the process of revision, we found that its inclusion diluted the main message of the paper. Therefore, as already stated in the initial manuscript, we have decided to postpone a detailed discussion of the pre-MPT results to future work. Nonetheless, in the revised manuscript, we have expanded the discussion on the potential extension of the timescale matching hypothesis to the 41-kyr world (lines 304–324). We believe this approach strikes an appropriate balance between maintaining the focus and enhancing the comprehensiveness of the manuscript.

*The distinction between nonlinear resonance and synchronization is well-explained, but the manuscript could better highlight why this distinction matters for the ∼100-kyr problem. For example, does the dominance of one mechanism over the other have implications for predicting future climate variability?*

The concepts of nonlinear resonance and synchronization underlie our timescale matching hypothesis. Therefore, a clear explanation of these concepts was necessary in this article. We have added the following sentences to explain the necessity of introducing these concepts in detail (Lines 42–44): **"Synchronization and nonlinear resonance are two major dynamical mechanisms that result in a system's response tightly coupled with external forcing. ... As they are central to the discussion that follows, we briefly review them below."**

On the other hand, the present study does not aim to determine which mechanism is the most plausible and thus the most suitable for long-term prediction. Rather, our objective was to provide a unified perspective on these mechanisms through the lens of the timescale matching problem. As you suggest, if one plausible mechanism is ultimately identified, it may have implications for predicting future climate variability. For example, if the Earth system exhibits self-sustained oscillations, it may have a tendency to enter a new ice age spontaneously over the next tens of thousands of years. We keep in mind that in reality, anthropogenic forcing plays a significant role, and any such prediction must clearly be verified using more realistic models.

*The power spectral density (PSD) analysis is robust, but the manuscript could include a more detailed comparison between model outputs and proxy records (e.g., time-domain metrics or phase*

[Figure]

Figure 1: Extension of the sensitivity experiments to the 41-kyr world before the MPT. (a) SO model corresponding to self-oscillatory dynamics. (b) SO model corresponding to resonant dynamics. (c) VCV18 model. (d) G23-3 model. Other descriptions are the same as Fig. 5 in the revised manuscript.

*relationships). This would help assess whether the models not only reproduce the ~100-kyr peak but also the timing of deglaciations.*

Yes, the time-domain metrics or phase relationships are definitely useful for assessing models with respect to the timings of deglaciations. The timings of deglaciations themselves are however complicated metrics because the last 800-kyr contains eleven deglaciations. The most simple time-domain metric would be the Pearson's correlation coefficient (PCC) between model outputs and the proxy record would. The PCC evaluates phase relationships, although it does not focus on deglaciations. In the revised manuscript, we have include a new Supplementary Fig. S5 demonstrating that timescale matching is also necessary to achieve a high correlation with the data, using the PCC as the metric. See lines 243–245: **"We note that the closeness between the intrinsic timescale and the 95-kyr eccentricity periodicity not only ensures the ~100-kyr dominant periodicity of ice age cycles but also enhances the temporal consistency between the simulations and the proxy data, as shown by the Pearson's correlation coefficients for varying parameters $r$ and $A$ in Fig. S5."** On the other hand, we cannot exclude the possibility that even good models may fail to reproduce the precise timing of deglaciations, particularly if that timing is sensitively dependent on parameters or influenced by stochastic forcings (Crucifix, 2013; Mitsui and Aihara, 2014; Mitsui et al., 2015; Mitsui and Crucifix, 2016). Therefore, caution is

warranted when using correlation-based metrics for model comparison.

*Figures S1–S7 are cited in the text but are not included in the preprint. The authors should ensure all supplementary figures are accessible or provide descriptions in the main text.*
It is unfortunate that you could not access the supplementary figures during the review. Actually they have been provided in the preprint page at: https://esd.copernicus.org/preprints/esd-2024-39/esd-2024-39-supplement.pdf. We will definitely upload the supplementary material according to Journal's submission guideline.

*Line 25: "Henceforth" should likely be "Previously."*
Indeed, "Henceforth" is not suitable in this context. Instead, we find that "hence" provides a clearer logical connection between the sentences. Accordingly, we have rephrased the text as follows: **Hence**, the ∼100-kyr glacial cycles have **previously** been explained as ...

*Lines 70-75: It only briefly explains each chapter's general content, not the research purpose and main methods, making it hard for readers to grasp the research core at the start. Suggest the author supplement research objective and main method info. When explaining objectives, state key scientific problems to solve and expected results. When describing methods, detail model selection criteria, simulation experiment process, and data analysis methods and ideas to help readers understand the paper's core content and research context.*
We acknowledge the reviewer's comment that the research purpose and main methods may not have been clearly conveyed. This could be partly because they were embedded mid-paragraph. Therefore, in the revised manuscript, we have changed the order of the paragraphs. Now the last two paragraphs of the Introduction to clearly state the objectives of the paper, outline the key scientific questions addressed, and describe the main methods, including model selection criteria, simulation procedures, and data analysis approaches. Please see lines 69-82.

*Line 204: The term "quasi-Arnold tongue" (Section 3.2) is introduced without a clear definition. A brief explanation or reference would aid readability.*
In the revised manuscript (lines 216–218), the quasi-Arnold tongues have been defined as **triangular regions where the principal frequency of a self-sustained oscillator under external forcing matches one of the forcing frequencies or a linear combination thereof.**

**Reply to Dr. Mikhail Verbitsky**

**CC1: Main point**

In the report by Dr. Verbitsky (CC1), he argues that *"astronomical forcing makes the intrinsic timescale irrelevant"* (p. 3) and hence *"the intrinsic timescale has no role in the present results"* (p. 3). We respectfully disagree with these statements. His argument is based on a scaling analysis. Using Buckingham's $\pi$-theorem, the system's response period $P$ is expressed as shown in his Eq. (8):

$$\frac{P}{\tau_{\text{int}}} = \Phi\left(\frac{\varepsilon}{a}, \frac{T}{\tau_{\text{int}}}, V\right), \tag{8}$$

where $\tau_{int}$ is the system's intrinsic timescale, $\varepsilon$ is the amplitude of the forcing, $a$ is the mass influx to the ice sheets, $T$ is a period of astronomical forcing, and $V$ is the parameter controlling the balance between positive and negative feedbacks. We agree with Eq. (8) itself, but he continues

with the assertion that *"we know from experiments with VCV model ... that for T =35–50 kyr ... the system responds with the period-doubling. This means that $\frac{P}{\tau_{int}}$ depends linearly on T, i.e.,*

$$\frac{P}{\tau_{int}} = \frac{T}{\tau_{int}}\Phi\left(\frac{\varepsilon}{a}, V\right), \;\; or \;\; \frac{P}{T} = \Phi\left(\frac{\varepsilon}{a}, V\right), \tag{9}$$

*we can see that astronomical forcing makes the intrinsic timescale irrelevant."* However, Eq. (9) is only locally true in the parameter space because $\frac{P}{\tau_{int}}$ depends nonlinearly on $\frac{T}{\tau_{int}}$ across different modes of resonances and non-resonances. This is shown in our Fig. 4b as a nonlinear dependence of $P$ on $\tau_{int}$ ($rT_0$ in our case). Therefore, we sustain our conclusion that the intrinsic timescale of the system is indeed critical for realizing the 100 kyr response. Of course, in each resonance mode, $P$ is fixed to $T$ or some combination of $T$s, consistently to his argument. Thus, in terms of $\pi$-theorem, Eq. (8) is a piecewise linear function of $T$, whose discontinuous points are determined by $\tau_{int}$.

In the revised manuscript, we clarified the above point in plain language (lines 247–251): **"Our sensitivity experiments show that the models' responses can lock into individual or combined astronomical frequencies, depending on their intrinsic timescales (Fig. 5). In particular, models tend to produce ∼100-kyr cycles when their intrinsic timescales are close to 100 kyr. This reflects a general property of synchronization and nonlinear resonance, observed across many ice age models (Table 1)."**

**Other confusions to be clarified**

- We have not stated that *"the system's response period to the astronomical forcing is independent of the amplitude of the astronomical forcing"* (p. 1 in his report). Actually, we have discussed the amplitude dependence of the phenomena in Section 3. To clarify this point, we have added the following sentence in the Discussion (lines 248–250): **"The locking frequency can also depend on the amplitude of the astronomical forcing (Figs 5b, c). However, under realistic forcing amplitudes, models tend to produce ∼100-kyr cycles when their intrinsic timescales are close to 100 kyr."**

- Dr. Verbitsky stresses that *"there is no similarity between ice sheets with and without forcing"* (his Section 4). We agree that ice-sheet dynamics with and without forcing are qualitatively different. Indeed, in the VCV18 model, the dynamics under weak forcing is close to a linear response to the obliquity cycles, while the dynamics under strong forcing is characterized by nonlinear resonance at ∼100-kyr time scales. Nevertheless, the result of sensitivity experiment shows that the response frequency is tightly coupled with the system's natural frequency in the absence of forcing (Fig. 5b). This is also consistent with the notion of resonance. Therefore, we conclude that the system's response to the forcing is tightly linked with the system's natural frequency in the absence of forcing. In the revised manuscript, we have clearly stated this point as follows (lines 229–234): **"Near the realistic forcing amplitude of $A \simeq 1$, resonances at 41 kyr, 95 kyr, and 124 kyr emerge when the scaled natural period $rT_0$ approaches these respective timescales (Figs. 5b, 6b). This correspondence indicates that resonance is driven by a timescale match between the system's natural period and an astronomical period, in line with the classical concept of resonance. We therefore conclude that the proximity between the system's intrinsic timescale and the 95-kyr eccentricity period is crucial for producing the 95-kyr cycles in the VCV18 model as well."**

- The natural period of the VCV18 model was calculated to be 95 kyr in our work. This value is coincidentally identical to the observed principal period of ice age cycles as well as one of the

eccentricity periods. However, we do not need this coincidence for our conclusion. The natural periodicity does not have to be sharply at 95 kyr for realizing the 95 kyr cycles. Indeed, the resonance at 95 kyr can occur for a range of natural periodicities, $83 \leq rT_0 \leq 118$ kyr for the realistic astronomical forcing. In the revised paper, we have mention as follows (Line 234–236): **"Note that the close numerical match between the natural period $T_0 = 95$ kyr and the 95 kyr eccentricity period is purely coincidental, and the resonance at 95 kyr can occur for a range of natural periods, $83 \leq rT_0 \leq 118$ kyr for the realistic astronomical forcing $A = 1$."**

We appreciate comments by Dr. Verbitsky again. A couple of equations introduced in his report help theoretical considerations. While his comments are critical, we believe that apparent contradictions between his opinions and our thoughts can be solved by careful revisions proposed above.

**Reply to report CC2**

In the report CC2, Dr. Verbitsky states that *"VCV18 bifurcation points can be described as a timescale matching problem between orbital timescale and orbitally modified intrinsic timescale"*. We agree with this point that the range of the intrinsic timescale allowing a particular resonance $(P \sim T)$ depends on the amplitude of astronomical forcing, and that such resonances are not observed if the forcing is too weak. In our reply **AC3**, we discussed a necessary condition for the resonance in terms of the ice accumulation rate. Although we believe the argument is thought-provoking, we find it not solid enough to include in the paper, as it lacks feedback from temperature and basal melting, which are key for the damped oscillation in the VCV18 model. Therefore, in the revised manuscript, we added the following clarification (lines 280–283): **"In contrast, in the nonlinear resonance mechanism with damped oscillations, the natural period leading to the ~100-kyr cycles can deviate from 100 kyr, depending on the forcing amplitude, as suggested by the tilted 95-kyr resonance region in Fig. 5b. Thus, this mechanism not require a precise match between the internal and external periods, but rather a general alignment of their timescales."**
* * *
**AC3**

On that basis, we point out that in many ice age models under astronomical forcing with realistic amplitude, ~100-kyr responses arise if the model's intrinsic timescale is close to ~100 kyr. That is, our conclusion holds for realistic forcing amplitude: $A \approx 1$ in our terminology and $\varepsilon \approx 1$ in VCV18's term.

Inspired by his scaling analysis, we propose the following physical argument. It does not use the Pi-theorem but as we show next it converges to a conclusion similar to his one. Following the VCV18 paper, the height of the fully developed ice sheet is given by $H = \zeta S_0^{1/4}$ and the snow accumulation rate is $a$. Using his comment (CC1), the intrinsic time scale of advection **in the absence of forcing** is given as

$$\tau_{adv} = \frac{H}{a} = \frac{\zeta S_0^{1/4}}{a}.$$

Since the snow accumulation rate $a$ and the forcing term $\varepsilon F_S(t)$ appear as $a - \varepsilon F_S(t)$ in the dynamical equations of VCV18, we assume that the **net ice accumulation rate under a forcing cycle** scales as $a - c\varepsilon$: this is similar to $a - \varepsilon F_S(t)$ but we introduce a cycle-specific coefficient $c$. Indeed, the astronomical forcing is a complicated signal and its amplitude from cycle to cycle. For

[Figure]

Figure 2: The parameter region derived from the simple physical consideration. The resonance with the astronomical period $T$ is possible at least within the triangular region.

this cycle (of period, say, $T$) to actually entertain a resonance with glaciation dynamics, we expect the typical ice build-up time to match $T$, i.e.,

$$\text{(glaciation period)} = \frac{\text{(maximal ice-sheet height)}}{\text{(net ice accumulation rate)}} = \frac{H}{a - c\varepsilon} \sim T.$$

$$\Leftrightarrow \quad c \sim \frac{1}{\varepsilon}\left(a - \frac{H}{T}\right) = \frac{a}{\varepsilon T}\left(T - \tau_{adv}\right).$$

This equation must hold for a majority of cycles, that is, for a range of $c$ denoted by $-c_1 < c < c_2$ ($c_1$, $c_2 > 0$). Thus,

$$-c_1 \lesssim \frac{a}{\varepsilon T}\left(T - \tau_{adv}\right) \lesssim c_2$$

That is,

$$\varepsilon \gtrsim \frac{a}{c_1 T}\left(\tau_{adv} - T\right) \quad \text{and} \quad \varepsilon \gtrsim \frac{a}{c_2 T}\left(T - \tau_{adv}\right).$$

These inequalities imply a triangular region in $\tau_{adv}$–$\varepsilon$ space (Fig. 1 here). The system may resonate at the astronomical period $T$ at least within the triangular region. If we interpret $\tau_{adv}$ as the intrinsic timescale of the system and if use the notation of our article ($\tau_{adv} = rT_0$ and $\varepsilon = A$), the above inequalities are

$$A \gtrsim \frac{a}{c_1 T}\left(rT_0 - T\right) \quad \text{and} \quad A \gtrsim \frac{a}{c_2 T}\left(T - rT_0\right).$$

The resonance may occur at least within this region, but the actual resonance region is more complicated than suggested from the above equations because of nonlinear effects (cf. Figs. 4 and S5 in our article).

The inequalities derived here are slightly different from what you drive using a scaling analysis in CC2. However, we reach essentially the same conclusion that the range of the intrinsic timescale leading to a particular resonance ($P \sim T$) must depend on the forcing amplitude if the forcing amplitude changes significantly. Our conclusion holds for the astronomical forcing with realistic amplitude. **This point will be addressed in the revised manuscript.**

We would like to thank you again for guiding us to the physical considerations.

**References**

Michel Crucifix. Why could ice ages be unpredictable? *Climate of the Past*, 9(5):2253–2267, 2013.

Andrey Ganopolski. Toward generalized milankovitch theory (gmt). *Climate of the Past*, 20:151–185, 2024.

Takahito Mitsui and Kazuyuki Aihara. Dynamics between order and chaos in conceptual models of glacial cycles. *Climate dynamics*, 42(11-12):3087–3099, 2014.

Takahito Mitsui and Michel Crucifix. Effects of additive noise on the stability of glacial cycles. *Mathematical Paradigms of Climate Science*, pages 93–113, 2016.

Takahito Mitsui, Michel Crucifix, and Kazuyuki Aihara. Bifurcations and strange nonchaotic attractors in a phase oscillator model of glacial–interglacial cycles. *Physica D: Nonlinear Phenomena*, 306:25–33, 2015.

Takahito Mitsui, Matteo Willeit, and Niklas Boers. Synchronization phenomena observed in glacial-interglacial cycles simulated in an earth system model of intermediate complexity. *Earth System Dynamics*, 2023:1277–1294, 2023. doi: 10.5194/esd-14-1277-2023. URL `https://doi.org/10.5194/esd-14-1277-2023`.

J Oerlemans. Glacial cycles and ice-sheet modelling. *Climatic Change*, 4(4):353–374, 1982.